# Opt-Verifier: Unleashing the Power of LLMs for Optimization Modeling via Dual-Side Verification

**Haoyang Liu** [1]   **Jie Wang** [1†]   **Boxuan Niu** [1]   **Xiongwei Han** [2]   **Yian Xu** [1]   **Mingxuan Ye** [1]   **Zijie Geng** [1]
**Fangzhou Zhu** [2]   **Tao Zhong** [2]   **Mingxuan Yuan** [2]   **Jianye Hao** [2 3]

## Abstract

Building mathematical optimization models is critical in operations research (OR), while it requires substantial human expertise. Recent advancements have utilized large language models (LLMs) to automate this modeling process. However, existing works often struggle to verify the correctness of the generated optimization models, without checking the rationality of the constraints and variables or the validity of solutions to the generated models. This hampers the subsequent verification and correction steps, and thus it severely hurts the modeling accuracy. To address this challenge, we propose a novel LLM-based framework with Dual-side Verification (Opt-Verifier) from both structure and solution perspectives, thereby improving the modeling accuracy. The structure-side verification ensures that the modeling structure of the generated optimization models aligns with the original problem description, accurately capturing the problem's constraints and requirements. Meanwhile, the solution-side verification interprets and evaluates the solutions' validity, confirming that the optimization models are logically and mathematically sound. Experiments on popular benchmarks demonstrate that our approach achieves over 20% improvement in accuracy.

## 1. Introduction

Optimization problems are foundational to operations research (OR), with wide applications in manufacturing (Jayal

---

This work was completed while Haoyang Liu <dgyoung@mail.ustc.edu.cn> was an intern at Huawei Technologies. [1] MoE Key Laboratory of Brain-inspired Intelligent Perception and Cognition, University of Science and Technology of China [2] Noah's Ark Lab, Huawei Technologies [3] Tianjin University . Correspondence to: Jie Wang <jiewangx@ustc.edu.cn>.

*Proceedings of the 43$^{rd}$ International Conference on Machine Learning*, Seoul, South Korea. PMLR 306, 2026. Copyright 2026 by the author(s).

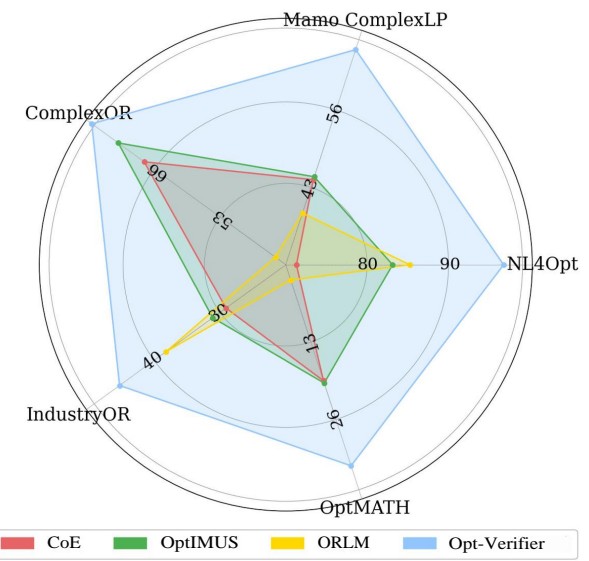

*Figure 1.* Opt-Verifier outperforms other baselines in solving accuracy (SA) across the benchmarks.

et al., 2010), transportation (Yin, 2002), and service industries (Berman et al., 1994). In practice, OR problem statements are typically specified in natural language. Practitioners must therefore (i) translate these descriptions into an appropriate mathematical optimization model (defining objectives, decision variables, and constraints) and (ii) implement the solver code (e.g., SCIP (Achterberg, 2009), Gurobi (Gurobi Optimization, 2021), or Pyomo (Bynum et al., 2021; Hart et al., 2011)) to obtain solutions. This workflow is labor-intensive, demands substantial domain expertise in problem context, mathematical modeling, and code-level implementation or debugging, and is consequently costly and time-consuming (Ahmaditeshnizi et al., 2024).

Given the impressive capabilities of large language models (LLMs) in natural-language understanding and domain knowledge acquisition, a growing number of works have employed LLMs to automate the processes of modeling, programming, and debugging. Existing approaches can be broadly grouped into two categories. The first category, prompt-based methods, relies on pre-trained LLMs (e.g., GPT-4 (OpenAI, 2023) and GPT-4o (OpenAI, 2024)), which

are prompted to construct mathematical models incrementally. In practice, these methods are often implemented into a carefully designed framework, such as multi-agent cooperation (Xiao et al., 2024; Ahmaditeshnizi et al., 2024) and Monte Carlo tree search (Astorga et al., 2025). The second category enhances modeling capabilities through fine-tuning, which involves constructing a large, labeled dataset for LLM training (Huang et al., 2025; Wu et al., 2025; Jiang et al., 2025; Chen et al., 2026; Lu et al., 2025).

Beyond these two approaches, recent research has explored self-correction strategies to improve the modeling performance (Jiang et al., 2025; Xiao et al., 2024; Ahmaditeshnizi et al., 2024). These methods trigger correction primarily from error messages produced during code execution. However, such strategies confine self-correction to code-level issues, and the underlying model can remain flawed even when the code runs without failure. To develop more effective model verification methods, we characterizes incorrect models by the following features. First, incorrect models often overlook indispensable constraints that the textual problem description does not explicitly state. For example, when formulating a maximum flow problem, an LLM might omit flow-balance constraints at intermediate nodes simply because they are not explicitly mentioned, yielding a structurally incomplete model. Second, incorrect models can yield solutions that are infeasible or violate basic logical principles, even though the solver executes successfully and reports an improved objective.

To address these challenges, we propose a novel multi-agent framework with Dual-side Verification (Opt-Verifier) from both the modeling structure and solution perspectives, improving the modeling accuracy. This approach moves beyond simple code-execution signals by translating both the optimization model and its resulting solutions into natural language and evaluating their semantic correctness. To the end, Opt-Verifier introduces two novel evaluation metrics for self-correction: modeling structure consistency and solution validity. (1) **Consistency in the modeling structure** ensures that the mathematical formulation (its variables, constraints, and objective) is a complete and faithful translation of the original problem description. To evaluate this metric, one LLM agent performs a back-translation that abstracts the generated model into a compact, multi-level description of its components. A second agent then aligns this abstraction with the structure derived from the original specification to reveal omissions or mismatches. (2) **Solution validity** assesses whether Othe solution of the produced model is logically and contextually sound for the real-world task. To assess it, one agent interprets the numeric solution in natural language, explaining its meaning in the context of the real-world problem. Another agent then critiques that interpretation to expose logical absurdities or mathematical violations that code-execution checks miss. Finally, we use the verification feedback for model refinement.

As illustrated in Figure 1, extensive experiments on five popular benchmarks showcase that our approach significantly outperforms the state-of-the-art, achieving an average improvement of approximately 10% in solving accuracy. Notably, Opt-Verifier is designed as a plug-and-play framework, capable of effectively verifying and refining optimization models generated by any existing pre-trained or fine-tuned OR LLMs.

## 2. Related Work

**Automated Optimization modeling** Machine Learning for solving Operations Research (OR) has drawn much attention in recent years (Bengio et al., 2021; Pu et al., 2026; Liu et al., 2025). In practice, the OR problems often arise from real-world situations, which are typically described in natural language. Consequently, automated optimization modeling has emerged as a critical area aimed at reducing the labor and time costs associated with the modeling process (Chen et al., 2023; Li et al., 2023). Notable early efforts in this field include the NL4Opt competition (Ramamonjison et al., 2021). Since then, several benchmarks have been introduced to evaluate performance, such as ComplexOR (Xiao et al., 2024), NLP4LP (Ahmaditeshnizi et al., 2024), Mamo (Huang et al., 2024), IndustryOR (Huang et al., 2025) and Optibench (Yang et al., 2025; Wang et al., 2024). Recent research primarily falls into two categories: prompt-based methods and fine-tuned methods. Prompt-based approaches utilize pre-trained large language models (LLMs) with carefully crafted prompts to iteratively construct models. The early works utilize the multi-agent techniques (Mao et al., 2026; Zhao et al., 2026; Zhang et al., 2026; Mao et al., 2026). For instance, Chain-of-Experts (Xiao et al., 2024) and OptiMUS (Ahmaditeshnizi et al., 2024) frameworks employ multi-agent cooperation, while some other methods (Astorga et al., 2025) leverage Monte Carlo tree search techniques to explore potential models. OptiTree (Liu et al., 2025) uses tree search to retrieve the LLM-collected modeling knowledge for improved performance, and Opt-Miner (Liu et al., 2026a) leverages web search to bridge the knowledge gap in semantic understanding and modeling techniques. To further enhance the modeling capabilities of LLMs, researchers also work on fine-tuning these models with extensive OR and modeling knowledge (Huang et al., 2025; Wu et al., 2025; Jiang et al., 2025; Chen et al., 2026; Liu et al., 2026b). For example, LLaMoCo (Ma et al., 2024) utilizes an instruction tuning framework to adapt LLMs for solving optimization problems in a code-to-code manner. ORLM (Huang et al., 2025) trains open-source LLMs specifically designed for optimization modeling and solver code development. Additionally, advanced techniques such as DPO (Rafailov et al., 2023; Ethayarajh et al., 2024; Liang

et al., 2025), GRPO (Shao et al., 2024) and data augmentation have been introduced to improve model training (Wu et al., 2025; Jiang et al., 2025).

# 3. Motivated Results and Case Analysis

**Challenges** We identify two key challenges in existing LLM-based optimization modeling methods. (1) LLMs struggle to identify the modeling structure within the problems, including missing or incorrect constraints and errors in variable definitions. The prevalence of such mistakes is notable in benchmarks, with 36.0% in NL4Opt and 12.6% in ComplexOR as pointed out in OptiMUS (Ahmaditeshnizi et al., 2024). (2) LLMs can only find the errors in the solver codes, but can hardly find the errors in the optimization models. OptiMUS points out that "Coding errors are easier to identify and fix. In contrast, identifying bugs in the formulation requires deeper reasoning and is harder." In existing methods, the debugging module is typically activated only when solver codes produce execution errors. To illustrate these challenges, we use GPT4o-mini in Figure 2, involving a maximum flow problem (MF).

**Observations on the Modeling Structures** To specify the definition and the usage of modeling structures, we have the following observation. The LLM cannot find the flow balance constraints at first. However, the model can correctly identify the relevant problem classifications. When we prompt the model to formulate the relevant problem classification (Maximum Flow Problem in this case), it successfully identifies the flow balance constraints.

The core principle of structure-augmented modeling is to **leverage similar standard optimization models as a reference** to identify a problem's implicit constraints. In Operations Research, many problems in similar scenarios share characteristics with optimization models in conventional problem classifications, such as the Vehicle Routing Problem or the Maximum Flow Problem. These classic types have conventional mathematical formulations——including standard variables and assumed constraints——which this paper refers to as **modeling structures**. Even if a new problem does not neatly fit a standard problem classification, referencing the modeling structure of a similar, well-understood problem helps the LLM uncover these implicit relationships, which are often crucial for correct formulations.

# 4. Methodology

Our work investigates how the modeling process can be enhanced through effective verification methods on both the structural and solution sides. An overview of the framework is presented in Figure 3. We define multi-level modeling structures in Section 4.1, followed by a detailed explanation of structure-side verification in Section 4.2 and solution-side

verification using a multi-agent cooperation framework in Section 4.3. We first introduce some notations in this work as follows.

Let $\mathcal{D}$ represent the space of natural problem descriptions, and let $\mathcal{M}$ denote the model space encompassing all possible optimization models. The modeling process can be viewed as a mapping from the problem description $D \in \mathcal{D}$ to an optimization model $M \in \mathcal{M}$.

## 4.1. Structure-Augmented Modeling

**(1) Motivation of Multi-Level Structure: Coarse-to-fine structure Analysis** Before the modeling process, human experts first analyze the problem description to identify a similar conventional problem classification as a reference. Next, they determine the variant of the classification that best aligns with the description. Finally, they assess special requirements in the description. This analysis follows a coarse-to-fine approach, from high-level to low-level structure analysis.

**(2) Multi-Level Modeling Structure** Our framework begins by distilling the modeling structures from the natural language descriptions. As discussed in Section 3, understanding the problem type is crucial in the modeling process, as it serves as a foundational template for developing optimization models. Inspired by the coarse-to-fine structure analysis process used by human experts, we further refine the concept of modeling structures by introducing the idea of multi-level modeling structures.

- High-Level Structure: This represents the fundamental problem type within OR, such as the maximum flow problem, set covering problem, vehicle routing problem, and knapsack problem. Each of these problem types is associated with a basic optimization model.

- Medium-Level Structure: This pertains to the classical classification or variants of fundamental problem types. For instance, variations of the maximum flow problem include multi-source MF, multi-commodity MF, minimum-cost MF, and MF in undirected graphs. Each variant is associated with a specific modified optimization model derived from the basic model.

- Lower-Level Structure: This level encompasses constraints in the classical optimization model as well as specific requirements that extend beyond classical models. For instance, standard constraints might include capacities and flow balance constraints in the MF, while special requirements could involve flow capacities that fluctuate over time.

As we mentioned in Section 3, even highly complex and unique industrial problems are often variants or combina-

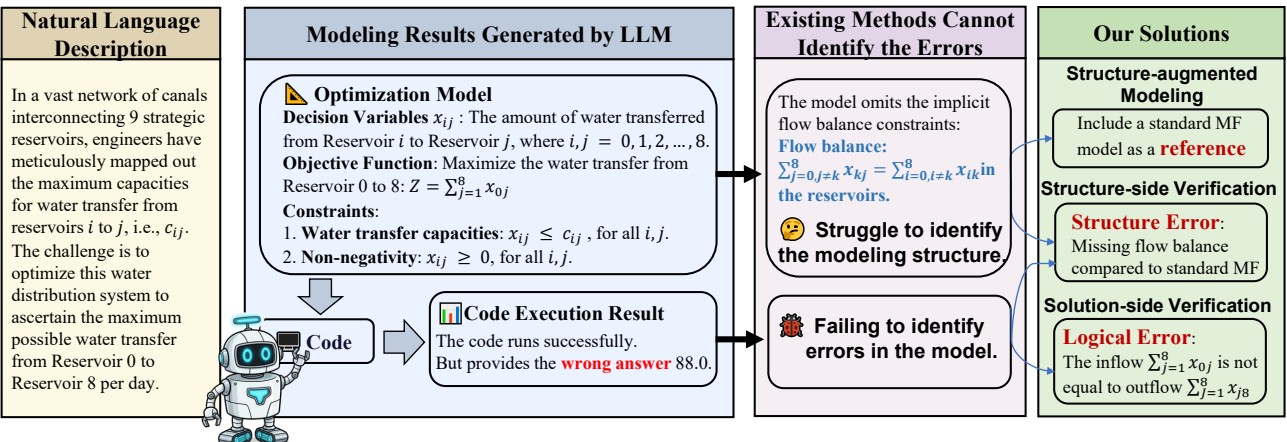

*Figure 2.* The two challenges we observed in existing optimization modeling methods.

tions of fundamental problem classifications recognized in operations research. The **high-level** and **medium-level structures** in our framework are designed to capture this foundational core, providing a solid starting point for the modeling process. Second, and most critically, the framework's **low-level structure** is specifically designed to provide the necessary flexibility to handle unique, real-world contexts. This level is not confined to a specific problem classification and is intended to capture the nuanced, problem-specific constraints and requirements that extend beyond classical formulations. This design allows the framework to represent the unique aspects of any given problem, rather than forcing it into a rigid, predefined category. We denote the multi-level modeling structure as $S$. We provide an example of the modeling structure we have defined.

---

**Example: The Structure Schema Extracted by LLMs**

- **High Level**: Maximum flow problem

- **Medium Level**: Single commodity maximum flow

- **Low Level**:

  - **Directed Network**: The flow is directed from one reservoir to another.
  - **Capacity Constraints**: Each edge (connection between reservoirs) has a maximum capacity.
  - **Flow Conservation**: The amount of water entering any intermediate reservoir must equal the amount leaving, except for the source and sink.

---

**(3) Structure Distillation and Structure-Augmented Modeling** We use two pre-trained LLMs (implemented by GPT4o-mini in this work) as agents to complete the structure distillation and initial modeling tasks, guided by designed prompts. To distill the multi-level modeling structure

$S$ from the natural language description $D$, we introduce an LLM agent, called the structure distillation agent. The agent takes as input the problem description and outputs the formatted structure context. Then, we call a formulation agent to generate an initial optimization model $M$ guided by prompts combining the problem description and modeling structure, i.e.,

$$S = \texttt{Distillation\_Agent}(D), \quad (1)$$
$$M = \texttt{Formulation\_Agent}(D, S). \quad (2)$$

### 4.2. Structure-Side: Structure Interpretation and Consistency Verification

**(1) Motivation** Structure-side verification finds the modeling errors by detecting any deviation from the established, correct formulation for a known class of problems, catching errors of omission where the LLM may overlook fundamental constraints and variables required for that problem classification. Inspired by dual learning in machine translation (He et al., 2016), we assert that a correct model must meet the following consistency criterion: when we translate the optimization model back into the space of modeling structure, the resulting context should semantically correspond to the modeling structure directly derived from the problem description.

**(2) Structure Interpretation and Consistency Verification** We introduce a structure interpretation agent and an evaluation agent to complete the structure verification task. The two agents are also guided with specific prompts. First, a structural interpretation agent performs a "back-translation". It takes the generated mathematical model $M$ and converts it back into its abstract modeling structure $\tilde{S}$. Next, a structural evaluation agent acts as a critic. It compares the interpretation agent's output $\tilde{S}$ with the original structure $S$ derived from the problem description to check for semantic

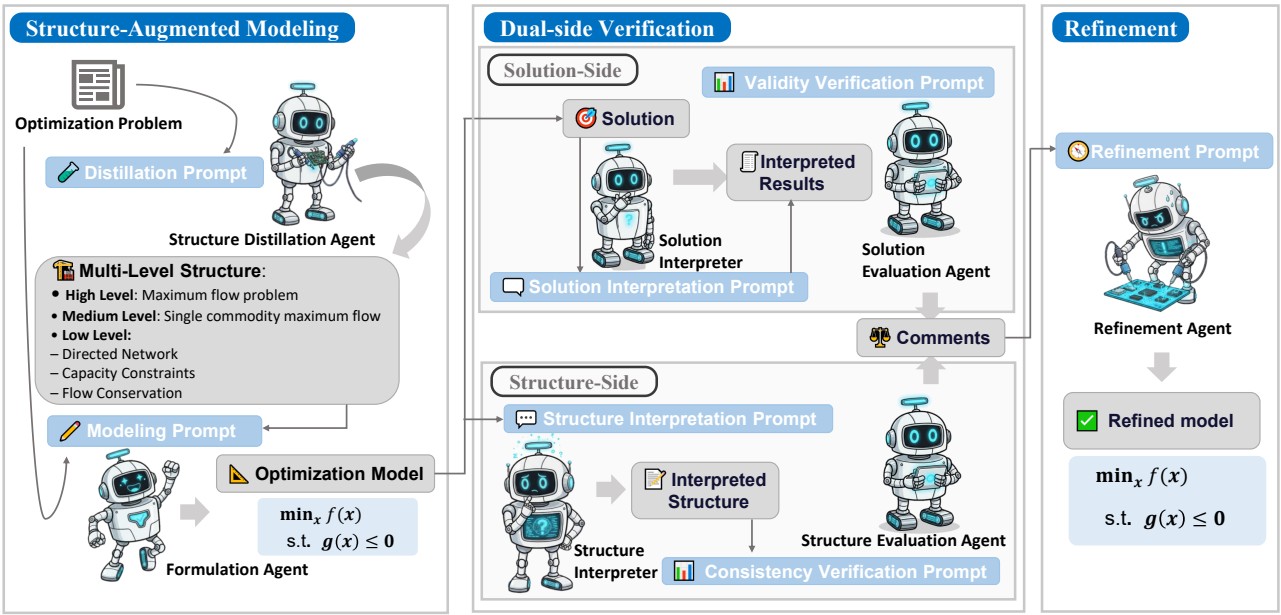

*Figure 3.* Our Opt-Verifier framework begins by distilling the multi-level structures from the natural language description. These extracted structures are then combined, allowing the formulator to generate an initial model. Then, Opt-Verifier conducts a dual-side verification and refinement process.

consistency. The evaluation agent's output is twofold: a binary consistency score $c_c$ and a detailed comment that highlights any discrepancies. This comment provides specific, actionable feedback that is later used to refine the model. The process can be formally summarized as

$$\tilde{S} = \texttt{StruInterp\_Agent}(M), \quad (3)$$

$$(com, c_c) = \texttt{StruEval\_Agent}(S, \tilde{S}), \quad (4)$$

where $c_c = 1$ indicates consistency, while the comment *com* details any differences found between the structures, guiding the subsequent refinement step.

Finally, we propose an analysis of the structure-side verification using mutual information. Suppose that $\tilde{S}$ is the interpreted structure from optimization model $M$.

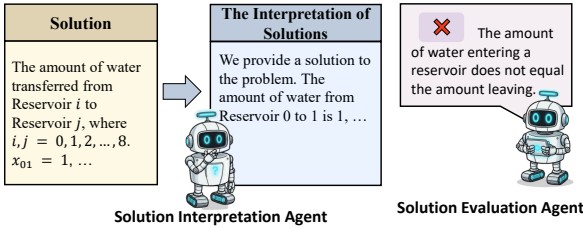

*Figure 4.* An example of solution verification.

## 4.3. Solution-Side: Solution Interpretation and Validity Verification

**(1) Motivation** This method works because it grounds the abstract mathematical model by assessing whether its solution is logically feasible. A model may be syntactically correct and yield a numerical answer, yet that answer could violate the fundamental logic of the original problem (e.g., suggesting more water flows out of a reservoir than flows in). We argue that the semantic content of the solution itself is a far richer source for identifying errors. Solution-side verification enhances performance because it is designed to catch logical errors that are invisible to systems that only check for solver execution errors. The core of this verification is to leverage the **common-sense and logical reasoning capabilities** of such LLMs for improved error detection.

**(2) Solution Interpretation and Validity Verification** Given an optimization model $M$, Opt-Verifier executes the solver code and obtains the optimal solution $x$. Then, Opt-Verifier performs solution-side

verification using two LLM agents, guided by designed prompts. The first agent, a solution interpreter, translates the raw numerical solution $x$ into a meaningful natural language description $\tilde{D}$ based on the original problem context $D$. Next, the second agent, a solution evaluation agent, scrutinizes this description to identify any logical or mathematical errors. This agent's output includes a binary validity

*Table 1.* Comparison of our method and the baselines across five popular benchmarks. Throughout the experiments, we compare the solving accuracy (SA) of the methods.

| | NL4Opt | Mamo ComplexLP | ComplexOR | IndustryOR | OptMATH | Micro Average |
|---|---|---|---|---|---|---|
| **Reasoning LLMs** | | | | | | |
| DeepSeek-R1 | 82.6 | 67.2 | 68.4 | 32.0 | 33.1 | 56.7 |
| OpenAI-o1 | 87.1 | 66.3 | 68.4 | 36.0 | 32.5 | 58.1 |
| **Fine-tuned Method** | | | | | | |
| ORLM | 85.1* | 38.8* | 42.1* | 38.0* | 2.6* | 41.3 |
| Evo-Step | 84.4* | 61.6* | - | 36.3* | - | - |
| LLMOPT | 80.3* | 44.1* | 72.7* | 29.0* | 12.5* | 47.7 |
| OptMATH | 95.9* | 54.1* | - | - | 34.9* | - |
| SIRL | 96.3* | 62.1* | - | 33.0* | 29.0* | - |
| **Prompt-based Method** | | | | | | |
| Standard | 64.6 | 27.9 | 31.5 | 24.0 | 15.6 | 32.7 |
| CoT | 69.3 | 34.5 | 36.8 | 27.0 | 18.6 | 37.2 |
| CoE | 71.3 | 44.5 | 68.4 | 29.0 | 19.8 | 46.6 |
| OptiMUS | 83.0 | 45.0 | 73.6 | 31.0 | 20.2 | 50.6 |
| Opt-Verifier (Ours) | **96.5** | **66.7** | **78.9** | **45.0** | **34.3** | 64.3 |

Values with * are from the original or reproduced papers, and - are with missing data because the model has not been publicly released.

score $c_v$ and, crucially, a detailed comment $com$ that provides specific feedback on any flaws found. The score value is 1 if the evaluator recognizes the validity of solution $x$, and 0 otherwise. This verification process can be formally summarized as:

$$\tilde{D} = \texttt{SolInterp\_Agent}(x, D), \quad (5)$$

$$(com, c_v) = \texttt{SolEval\_Agent}(D, \tilde{D}). \quad (6)$$

Similar to the analysis of the structure-side verification, we have the following analysis of the solution-side verification. Suppose that $\tilde{D}$ is the interpreted solution from the optimization model $M$.

### 4.4. Refinement

Based on the feedback, we refine the optimization model. The refinement agent within the Opt-Verifier framework is a specialized LLM-based component responsible for correcting and enhancing the initial optimization model based on insights from the dual-side verification process. Guided by a refinement prompt, the agent takes the current formulation as input and produces a refined optimization model, represented as $M' = \texttt{Ref\_Agent}(D, S, M, com)$.

## 5. Experiments

**Benchmarks** We use five real-world operations research benchmarks: NL4Opt (Ramamonjison et al., 2021), Mamo ComplexLP (Huang et al., 2024), ComplexOR (Xiao et al., 2024), IndustryOR (Huang et al., 2025) and OptMATH (Lu et al., 2025). The NL4Opt benchmark, released for the NeurIPS 2022 NL4Opt competition, consists of 289 elemen-

tary linear programming problems. Mamo ComplexLP 211 problems. ComplexOR is a comprehensive dataset including linear and mixed-integer programming. In alignment with the studies by (Ahmaditeshnizi et al., 2024) and (Jiang et al., 2025), we focus on 19 specific problems from this dataset. IndustryOR has 100 problems from various industry scenarios. OptMATH has 166 challenging problems.

**Implementation and Baselines** In our experiments, we utilized the GPT4o-mini to implement the agents in our method and all the prompt-based baselines. For the implementation of Opt-Verifier, please see Appendix G for the prompts of each agent. In our experiments, we compare Opt-Verifier with four available prompt-based methods and five fine-tuned operations research LLMs. The four prompt-based baselines include Standard, Chain-of-Thoughts (CoT) (Wei et al., 2022), Chain-of-Experts (CoE) (Xiao et al., 2024), and OptiMUS (Ahmaditeshnizi et al., 2024). The Standard baseline represents the output of GPT without any optimization of its reasoning processes. We include five fine-tuned open-source operations research language models as baselines, including ORLM (Huang et al., 2025) (based on LLaMA-3-8B model), Evo-Step (Wu et al., 2025) (based on LLaMA-3-8B model), LLMOPT (Jiang et al., 2025) (based on Qwen1.5-14B), OptMATH (Lu et al., 2025) (based on Qwen2.5-32B), and SIRL (Chen et al., 2026) (based on Qwen2.5-7B) trained with reinforcement learning. Additionally, we also compare our results with the pre-trained reasoning model DeepSeek-R1 (DeepSeek-AI, 2025) and OpenAI-o1 (OpenAI, 2024). For further advanced baselines, we also compare with OptiMind and (Zhang et al., 2026) AlphaOpt (Kong et al., 2026) in Appendix A.7

*Table 2.* Alation studies on (1) each component and (2) each level of modeling structures in Opt-Verifier.

| Method | NL4Opt | Mamo ComplexLP | ComplexOR | IndustryOR |
|---|---|---|---|---|
| Ablation for the Components | | | | |
| Opt-Verifier w/o struaug | 91.8 | 54.2 | 63.1 | 34.0 |
| Opt-Verifier w/o stru-side | 91.5 | 55.2 | 68.4 | 29.0 |
| Opt-Verifier w/o sol-side | 91.8 | 53.1 | 68.4 | 41.0 |
| Ablation for Each Level of the Structure | | | | |
| Opt-Verifier w/o high | 92.9 | 59.9 | 78.9 | 41.0 |
| Opt-Verifier w/o medium | 91.1 | 57.0 | 73.6 | 38.0 |
| Opt-Verifier w/o low | 91.1 | 54.9 | 73.6 | 30.0 |
| **Opt-Verifier (full)** | **96.5** | **66.7** | **78.9** | **45.0** |

**Metrics**  Consistent with existing research, we employed solving accuracy (SA) to evaluate performance. Specifically, SA represents the proportion of problems for which the methods successfully identify the optimal solutions. The higher value of SA implies better performance.

## 5.1. Main Results

To demonstrate the effectiveness of our method, we conduct experiments comparing solving accuracy (SA) between our approach and baseline across various benchmarks. The results presented in Table 1 indicate that our method significantly outperforms the baselines, achieving an approximate 20% improvement in solving accuracy compared to Standard. For the challenging benchmarks, our method consistently delivers outstanding performance. This demonstrates that Opt-Verifier exhibits strong generalization capabilities across both easy and difficult scenarios. Furthermore, Opt-Verifier achieves performance better than state-of-the-art reasoning LLMs, such as DeepSeek-R1 and OpenAI-o1, despite relying on a much weaker base model, GPT4o-mini. Please see Appendices E and F for case and error analysis.

## 5.2. Ablation Studies

**(1) The Effects of Each Component of Opt-Verifier**  In this section, we examine the effects of the three components of Opt-Verifier: structure-augmented modeling, structure-side verification, and solution-side verification. To assess their contributions, we implement three variants of Opt-Verifier. The first variant, Opt-Verifier w/o stru-aug, omits the introduction of a modeling structure to enhance the modeling process. For structure-side verification, instead of interpreting the model in structural terms, we instruct an LLM agent to provide a narrative explaining the meaning of the variables, constraints, and objectives. The second variant, Opt-Verifier w/o stru-side, does not implement structure-side verification at all. The third variant, Opt-Verifier w/o sol-side, excludes the solution-side verification process. The

results, presented in Table 2, reveal a significant drop in performance in the absence of any of these components, highlighting their essential roles in the modeling process.

**(2) The Effects of Each Level of Modeling Structures**  Next, we investigate the impact of each level within our proposed modeling structure. The variant Opt-Verifier w/o high/medium/low level excludes the use of high, medium, and low-level structures. The experimental results in Table 2 demonstrate that all three levels contribute positively to overall performance, with the medium and low-level structures showing particularly pronounced improvements.

**Takeaway**  Critically, the framework's "low-level structure" is specifically designed to provide the necessary flexibility to handle unique, real-world contexts. This level is not confined to a specific problem type and is intended to capture the nuanced, problem-specific constraints and requirements that extend beyond classical formulations.

*Table 4.* Inference efficiency comparison.

| Metric | Method | NL4Opt | MAMO | IndustryOR |
|---|---|---|---|---|
| Time (s) | CoE | 58.2 | 72.5 | 79.9 |
| | OptiMUS | 64.2 | 80.3 | 88.2 |
| | Ours | **52.8** | **67.6** | **59.4** |
| Agent Calls | CoE | 13.6 | 15.7 | 14.1 |
| | OptiMUS | 10.4 | 13.8 | 13.9 |
| | Ours | **9.0** | **9.1** | **9.2** |
| Tokens | CoE | 6745.8 | 7705.0 | 8751.5 |
| | OptiMUS | 7039.4 | 8248.4 | 9062.7 |
| | Ours | **5320.7** | **7385.2** | **6819.8** |

## 5.3. Comparison of Solving Efficiency

We examine the solving efficiency of Opt-Verifier in comparison to the prompt-based baselines CoE and OptiMUS by analyzing the average time, token usage, and agent calls required to solve a problem. We use the same solver (Gurobi)

*Table 3.* We build Opt-Verifier on different baselines and backbone models.

| Method | NL4Opt | Mamo ComplexLP | ComplexOR | IndustryOR |
|---|---|---|---|---|
| Different Baselines | | | | |
| ORLM | 85.1 | 38.8 | 42.1 | 38.0 |
| ORLM+Opt-Verifier | 92.3 | 59.6 | 73.6 | 42.0 |
| OptiMUS | 83.0 | 45.0 | 73.6 | 31.0 |
| OptiMUS+Opt-Verifier | 96.1 | 61.0 | 78.9 | 45.0 |
| Different Backbones | | | | |
| GPT-4o | 79.4 | 45.0 | 57.8 | 27.0 |
| GPT-4o+Opt-Verifier | 97.5 | 66.3 | 78.9 | 48.0 |
| Qwen2.5-14B | 70.3 | 41.2 | 57.8 | 31.0 |
| Qwen2.5-14B+Opt-Verifier | 85.8 | 56.3 | 68.4 | 39.0 |

*Table 5.* Verification Precision

| Verification Type | **Easy** | **Medium** | **Hard** |
|---|---|---|---|
| Structure Verification | 92% | 89% | 83% |
| Solution Verification | 93% | 91% | 86% |

*Table 6.* Verification Recall

| Verification Type | **Easy** | **Medium** | **Hard** |
|---|---|---|---|
| Structure Verification | 86% | 79% | 68% |
| Solution Verification | 83% | 85% | 73% |

for all methods to ensure fairness. The solving time reported in Table 4 comprises the modeling time using LLMs and the execution time of the solver. The results presented in Table 4 indicate that Opt-Verifier achieves significantly shorter solving times, demonstrating high efficiency.

### 5.4. Building on Different Baselines and LLMs

**(1) Improving different Baselines:** Opt-Verifier was applied to the outputs of three foundational baselines: OptiMUS and the fine-tuned ORLM model. In each case, Opt-Verifier's verification and refinement process enhanced the initial models generated by these baseline methods. **(2)Improving different Base LLMs:** To illustrate that the framework is not reliant on a specific backbone model, we conducted experiments using various base LLMs with Opt-Verifier. This approach highlights how performance scales with the capabilities of the underlying model, including stronger models (e.g., GPT-4o) and weaker models (e.g., Qwen2.5-14B) The results consistently indicated significant performance gains, as shown in Table 3. Please refer to Appendix B for detailed experiment settings.

### 5.5. Quantity Analysis of Verifications

**Critical Components of Verifications** The interpretation and evaluation agents are essential components of the ver-

ification process, as they determine whether Opt-Verifier can effectively identify errors in the modeling process. We conducted extensive ablation studies to quantitatively assess the accuracy and reliability of these agents. Our experiments were specifically designed to evaluate their ability to distinguish between correct and incorrect models.

**Experiment design** We utilized the IndustryOR dataset for evaluation, which consists of three difficulty levels (easy, medium, and hard) that allow us to test the generalization capabilities of Opt-Verifier across varying problem complexities. *The hard problems can be general problems with complex structures that fall out of the conventional problem classifications.* However, this analysis was labor-intensive, as the IndustryOR dataset does not provide detailed, step-by-step ground-truth labels necessary for our analysis. To ensure the correctness of this evaluation, we resorted to a manual checking process, which is time-consuming. We first manually annotated the optimization models for the sampled problems to establish a ground truth. To facilitate our evaluation, we randomly selected ten problems from each difficulty level. For generating incorrect models, we initially labeled the models and extracted the structures. We then created nine negative samples for each labeled model by randomly deleting or rewriting some of the variables and constraints. This resulted in 30 positive and 270 negative modeling samples. For **structure evaluation**, we collected interpreted structures from both the positive and negative samples. The evaluator compared these interpreted structures with the ground-truth structures and generated a binary score. For **solution evaluation**, we used the positive and negative samples to generate solutions, which we then interpreted and assessed for reliability.

**Results** The evaluation accuracy and recall rates are presented in Tables 5 and 6. For each difficulty level, we evaluated 10 positive samples and 90 negative samples. Both precision and recall rates for the negative samples are high across the difficulty levels, demonstrating the reliability of

the scores. We find that the verification process still performs well for hard problems that cannot be classified into a specific problem type, indicating the strong generalization to general problems.

## 6. Conclusion

In this paper, we propose an LLM-based verification framework designed to enhance the accuracy of automated mathematical modeling tasks. In the structure-side verification, we assess the modeling structures of the current model to ensure structural consistency. Meanwhile, in the solution-side verification, we interpret the solution within the context of the problem descriptions, aiming to identify any logical or mathematical errors in the models. Extensive experiments demonstrate the effectiveness of our method across a wide range of benchmarks.

## Acknowlegement

This work was supported in part by the National Key R&D Program of China under contract 2022ZD0119801, National Nature Science Foundations of China grants U23A20388 and 62021001. We would like to thank all the anonymous reviewers for their insightful comments.

## Impact Statement

This paper presents work whose goal is to advance the field of machine learning. There are many potential societal consequences of our work, none of which we feel must be specifically highlighted here.

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

# A. Additional Experimental Results and Analysis

## A.1. Experiments on Cleaned Benchmarks

To ensure a fair and controlled comparison, we evaluate Opt-Verifier on the latest cleaned benchmark versions from the LLM4OR (Xiao et al., 2025) dataset. This address the concern regarding the consistency of benchmark versions. The results in Table 7 show that our approach consistently outperforms other baselines across all datasets. Notably, on the ReSocratic (Yang et al., 2025) benchmark, Opt-Verifier achieves a solving accuracy of 89.6%, which is significantly higher than the 48.4% achieved by the standard prompting method.

*Table 7.* Comparison of solving accuracy on the cleaned benchmarks.

| Method | NL4Opt | Mamo | ComplexOR | IndustryOR | ReSocratic |
|---|---|---|---|---|---|
| Standard | 61.2 | 57.7 | 42.9 | 38.1 | 48.4 |
| CoT | 62.2 | 42.3 | 39.2 | 40.5 | 43.6 |
| ORLM | 73.8 | 59.5 | 50.0 | 42.9 | 61.8 |
| OptiMUS | 76.2 | 46.8 | 46.8 | 45.2 | 87.6 |
| Ours | 86.0 | 78.3 | 61.1 | 52.3 | 89.6 |

## A.2. Stability Analysis and Confidence Intervals

To address the stochastic nature of LLMs, we conduct five independent trials for each benchmark to calculate the standard deviation. As shown in Table 8, the low standard deviation across all tasks demonstrates the stability of our framework. For instance, on the NL4Opt dataset, our method achieves an average accuracy of 96.5% with a standard deviation of only 0.4%.

*Table 8.* Solving accuracy with standard deviation across five trials.

| Method | NL4Opt | Mamo | ComplexOR | IndustryOR | OptMATH |
|---|---|---|---|---|---|
| Standard | 64.6 (0.5) | 27.9 (1.3) | 31.5 (1.3) | 24.0 (0.9) | 15.6 (0.4) |
| CoT | 69.3 (0.3) | 34.5 (1.6) | 36.8 (2.2) | 27.0 (1.1) | 18.6 (0.2) |
| OptiMUS | 83.0 (0.6) | 45.0 (1.1) | 73.6 (2.5) | 31.0 (0.8) | 20.2 (0.5) |
| Ours | 96.5 (0.4) | 66.7 (1.1) | 78.9 (0.0) | 45.0 (1.2) | 34.3 (0.4) |

## A.3. Reliability Analysis of Verifiers

We evaluate the reliability of our structure-side and solution-side verifiers by testing them on real LLM-generated modeling errors. We categorize these errors into variable, constraint, objective, and parameter errors. Table 9 reports the precision and recall for both verifiers. The results show that our dual-side verification mechanism can effectively identify various types of errors. For example, the solution-side verification achieves a recall of 0.95 for variable errors, confirming its ability to detect logical contradictions that simple code execution checks might miss.

*Table 9.* Reliability of verifiers on LLM-generated errors.

| Error Type | Structure-Side Verification | | Solution-Side Verification | |
|---|---|---|---|---|
| | Precision | Recall | Precision | Recall |
| Variable Errors | 0.88 | 0.83 | 0.92 | 0.95 |
| Constraint Errors | 0.75 | 0.67 | 0.80 | 0.79 |
| Objective Errors | 0.94 | 0.89 | 0.97 | 0.94 |
| Parameter Errors | - | - | 0.89 | 0.72 |

## A.4. Comparison of Solving Efficiency

**(1) Efficiency Definition**  We examine the solving efficiency of Opt-Verifier in comparison to the prompt-based baselines CoE and OptiMUS by analyzing the average time, token usage and agent calls to solve a problem. We use the **same solver (Gurobi)** for all methods. This ensures fairness in efficiency comparisons. The solving time in Table 10 contains the modeling time using LLMs and the execution time of the solver. The solver execution time is short (under 0.01 seconds) and

*Table 10.* Inference efficiency comparison of different methods.

| Metric | Method | NL4Opt | MAMO ComplexLP | ComplexOR | IndustryOR |
|---|---|---|---|---|---|
| Inference Time | CoE | 58.2 | 72.5 | 98.6 | 79.9 |
| | OptiMUS | 64.2 | 80.3 | 101.3 | 88.2 |
| | Ours | 52.8 | 67.6 | 68.2 | 59.4 |
| Agent Calls | CoE | 13.6 | 15.7 | 16.6 | 14.1 |
| | OptiMUS | 10.4 | 13.8 | 15.3 | 13.9 |
| | Ours | 9.0 | 9.1 | 9.4 | 9.2 |
| Token Usage | CoE | 6745.8 | 7705.0 | 9469.8 | 8751.5 |
| | OptiMUS | 7039.4 | 8248.4 | 10796.1 | 9062.7 |
| | Ours | 5320.7 | 7385.2 | 8457.4 | 6819.8 |

can be neglected during this process. Thus, the solving time in Table 10 reflects the modeling time by LLMs. The results presented in Table 10 indicate that Opt-Verifier achieves significantly shorter solving times, showcasing its high efficiency.

**(2) The Reason why Opt-Verifier is efficient**  Compared to other prompt-based baselines, **Opt-Verifier has a simpler workflow**. CoE and OptiMUS are based on the multi-agent cooperation framework. The workflow of these methods is automatically controlled by a management agent. The insufficient decision-making ability of the management agent may lead to suboptimal decision chains. In the experiments, we find that this method may repeatedly call the same agent. For example, for certain complex problems, CoE may call the terminology interpreter again and again. In contrast, DeVet does not include such a management agent, leading to a simpler workflow.

### A.5. Step-wise Efficiency Analysis

Although Opt-Verifier consists of multiple sequential steps, its overall computational cost remains low due to the design of each step and their interactions. In this subsection, we analyze efficiency from a step-wise perspective and explain why Opt-Verifier is more efficient in practice.

First, each step in Opt-Verifier is purpose-specific and executed exactly once. The workflow is fixed and directed, which prevents redundant reasoning and unnecessary re-invocation of agents. In contrast, existing multi-agent methods rely on a management agent to dynamically decide which agent to call next. Due to the stochastic nature of LLMs, this often results in repeated calls to the same agent and inefficient decision chains. By construction, Opt-Verifier avoids such loops and ensures that computation progresses monotonically toward a valid solution.

Second, the verification steps in Opt-Verifier are computationally lightweight. As shown in Table 11, the structure-side and solution-side verification steps consume significantly fewer tokens than the modeling and coding/debugging steps across all benchmarks. This indicates that verification introduces only a small overhead at the step level, rather than becoming a dominant cost factor.

Third, early error detection further improves efficiency at later steps. The verification steps are placed before refinement and coding/debugging, enabling Opt-Verifier to identify structural and logical errors at an early stage. This reduces the likelihood of costly downstream failures, such as repeated code debugging or solver re-runs, which are typically much more expensive in terms of tokens and time. Consequently, a small additional cost in verification leads to a net reduction in overall computation.

In summary, Opt-Verifier achieves high efficiency not by minimizing the number of steps, but by carefully designing each step to be non-redundant, lightweight, and strategically positioned within the workflow. This step-wise design allows Opt-Verifier to outperform existing multi-agent baselines in both computational efficiency and solution reliability.

*Table 11.* Average token usage of each stage in Opt-Verifier across benchmarks.

| Stage | NL4Opt | MAMO | ComplexOR | IndustryOR |
|---|---|---|---|---|
| Structure Distillation | 492.1 | 441.1 | 665.4 | 501.1 |
| Modeling | 1175.5 | 1944.8 | 2147.1 | 1564.7 |
| Structure-side Verification | 474.6 | 597.4 | 594.2 | 662.7 |
| Solution-side Verification | 302.4 | 422.1 | 521.3 | 347.4 |
| Refinement | 728.7 | 915.5 | 1013.5 | 759.0 |
| Coding and Debugging | 2147.4 | 3064.4 | 3515.9 | 2984.9 |

## A.6. The Problems We Try to Address is Critical in Optimization Modeling

The Opt-Verifier framework is designed and validated for broad applicability across a wide range of problem domains and model types. Our experimental results provide compelling evidence for this generalizability.

We demonstrate that the motivations and challenges we address are common and critical in the optimization modeling field.

*Table 12.* The proportion of errors.

|  | NL4Opt | Mamo ComplexLP |
|---|---|---|
| Missing Constraints | 37.3 | 20.0 |
| Failure Model Debugging | 51.8 | 40.0 |

**The missing constraints** Section 4.5 of the OptiMUS paper (Ahmaditeshnizi et al., 2024) has summarized and classified common errors, including missing or wrong constraints, incorrect model, and coding errors. Missing or wrong constraints mean the model fails to extract all the constraints from the model or generates wrong constraints. An incorrect model means errors, such as defining binary variables for visiting cities instead of links in TSP. The prevalence of such mistakes is notable in benchmarks, with 36.0% in NL4Opt and 12.6% in ComplexOR.

**Incorrect model debugging** This is also a common challenge. The OptiMUS paper (Ahmaditeshnizi et al., 2024) points out that "Coding errors are easier to identify and fix. In contrast, identifying bugs in the formulation requires deeper reasoning and is harder." In existing methods, the debugging module is called only when the solver codes raise execution errors.

## A.7. Comparison with OptiMind and AlphaOpt

We further compare Opt-Verifier with recent optimization modeling frameworks, namely OptiMind (Zhang et al., 2026) and AlphaOpt (Kong et al., 2026). These methods primarily focus on hint-based enhancement or iterative self-correction by collecting external experiences. However, they often rely on the model's intrinsic reasoning to identify errors, which can lead to a self-justification loop where the model confirms its own flawed logic.

In contrast, Opt-Verifier introduces a dual-side verification mechanism that provides dense correction feedback based on systematic mathematical and logical checks. Our approach is designed as a plug-and-play framework that can be integrated with these existing methods to further improve their performance.

As shown in Table 13, we evaluate the performance of OptiMind and AlphaOpt when enhanced by our verification module. The results demonstrate that Opt-Verifier serves as a highly effective enhancement. For instance, after integrating our verification module, the solving accuracy of AlphaOpt on the Mamo ComplexLP dataset increases from 62.6% to 68.7%, and OptiMind's accuracy on the OptMATH dataset improves from 33.7% to 34.9%. This highlights that the structural and solution-side feedback provided by Opt-Verifier captures errors that are often missed by hint-based or standard self-correction methods.

## B. Experiment Setting in Section 5.4

The plug-and-play capability of Opt-Verifier is supported by both its architectural design and empirical results across diverse setups. We have built Opt-Verifier based on two modeling baselines (OptiMUS and ORLM) and two pretrained

*Table 13.* Performance enhancement on different baseline frameworks.

| Method | NL4Opt | Mamo ComplexLP | IndustryOR | OptMATH |
|---|---|---|---|---|
| AlphaOpt (GPT-4o-mini) | 90.3 | 62.6 | 38.0 | 31.9 |
| AlphaOpt + Ours | 93.4 | 68.7 | 44.0 | 34.3 |
| OptiMind | 94.1 | 64.0 | 42.0 | 33.7 |
| OptiMind + Ours | 96.2 | 69.7 | 48.0 | 34.9 |
| Ours (GPT-4o-mini) | 96.5 | 66.7 | 45.0 | 34.3 |

backbone LLMs (GPT-4o and Qwen2.5-14B). OptiMUS are prompt-based methods with general LLMs as backbones (we use GPT4o-mini here), which can process any text inputs. We first extract multi-level structures for the problems using a structure distillation agent. These extracted structures are then appended to the problem descriptions and sent as input to the OptiMUS. Once the baselines generate an initial formulation, we proceed with Opt-Verifier's verification step. However, ORLM is a fine-tuned model designed to handle only specific input formats. Therefore, the ORLM model is used solely to provide an initial optimization model, while we perform the verification and refinement processes using the GPT4o-mini model.

## C. Ability of Handling Actual Large-Scale Problems

In this section, we use a toy problem to explain how Opt-Verifier is designed to handle real-world large-scale optimization problems and provide additional empirical evidence to support its scalability and reliability.

**Problem** (simplified version) The capacitated warehouse location problem aims to select a limited number of warehouses and allocate customer demands to them at minimum total cost. Each warehouse has a fixed operating cost and a finite service capacity, while each customer has a known demand that must be fully satisfied. Service allocation incurs transportation costs, and any opened warehouse must meet a minimum served demand. The objective is to minimize the sum of service allocation and warehouse operating costs subject to capacity and cardinality constraints.

### C.1. Abstract Parameterized Model Definition

Opt-Verifier generates parameterized optimization formulations rather than hardcoded, instance-specific models. The formulation agent produces an abstract mathematical and programmatic model in which the problem size is represented by symbolic parameters instead of explicit numerical values. For example, in a warehouse location problem, the formulation uses parameters such as `n_warehouses` and `n_customers`, and constraints are constructed through loops, e.g., `for j in range(n_customers): addConstr(...)`. As a result, the formulation logic is fully agnostic to problem scale. Whether the number of customers is 20 (as used in verification) or 20,000 (as in a real deployment), the same modeling logic applies without modification.

**Large-Scale Handling Process**

```python
# Decision Variables
# x[i,j] represents the amount of customer j's demand served by warehouse i
x = model.addVars(n_warehouses, n_customers, name="x", lb=0, vtype=GRB.CONTINUOUS)
# y[i] is binary: 1 if warehouse i is open, 0 otherwise.
y = model.addVars(n_warehouses, name="y", vtype=GRB.BINARY)

# Objective Function: Minimize total service allocation costs plus fixed operating
    ↪ costs for opened warehouses.
model.setObjective(
 gp.quicksum(service_allocation_cost[i][j] * x[i, j] for i in range(n_warehouses) for
     ↪  j in range(n_customers))
   + gp.quicksum(warehouse_fixed_cost[i] * y[i] for i in range(n_warehouses)),
   GRB.MINIMIZE
)

# Constraints

# 1. Demand Satisfaction: Each customer's entire demand must be met.
for j in range(n_customers):
   model.addConstr(gp.quicksum(x[i, j] for i in range(n_warehouses)) ==
       ↪ customer_demand[j],
              name=f"DemandCustomer_{j}")

# 2. Capacity Constraints and Linking x and y: Each warehouse cannot supply more than
    ↪  its capacity and if it is not open, no supply.
for i in range(n_warehouses):
   # Total supply from warehouse i should not exceed its capacity multiplied by y[i]
   model.addConstr(gp.quicksum(x[i, j] for j in range(n_customers)) <=
       ↪ warehouse_capacity[i] * y[i],
              name=f"CapacityWarehouse_{i}")
   # If warehouse i is open, it must meet at least a minimum demand.
   model.addConstr(gp.quicksum(x[i, j] for j in range(n_customers)) >=
       ↪ minimum_demand_from_warehouse[i] * y[i],
              name=f"MinDemandIfOpen_{i}")

# 3. Number of warehouses open: Must be at least the minimum and at most the maximum
    ↪ allowed.
model.addConstr(gp.quicksum(y[i] for i in range(n_warehouses)) >= min_open_warehouses,
    ↪  name="MinOpenWarehouses")
model.addConstr(gp.quicksum(y[i] for i in range(n_warehouses)) <= max_open_warehouses,
    ↪  name="MaxOpenWarehouses")
```

### C.2. Structure-side Verification of Parameterized Logic

The structure evaluation agent verifies the correctness of the modeling logic at an abstract level, independent of concrete parameter values. Its goal is to ensure that all required structural relationships are present in the formulation. For instance, it checks whether capacity constraints are correctly modeled and whether relationships such as

$$\sum_j x_{i,j} \leq \text{capacity}_i \cdot y_i$$

exist in the formulation. Importantly, the agent does not verify specific numerical values (e.g., whether $\text{capacity}_i$ equals 3,010 or 300,000), but instead confirms that the mathematical relationships between variables and parameters are logically correct. This design ensures that structural correctness generalizes naturally to large-scale instances.

## C.3. Solution-side Verification via Toy Instantiation

A common modeling error made by LLMs is the omission of binary decision variables in conditional constraints. For example, the LLM may fail to correctly link the warehouse opening variable $y_i$ to the capacity-related constraints, resulting in the following incorrect formulations:

$$\sum_{j \in [n_{\text{customers}}]} x_{ij} \leq \texttt{warehouse\_capacity}[i], \quad \forall i \in [n_{\text{warehouses}}], \tag{7}$$

$$\sum_{j \in [n_{\text{customers}}]} x_{ij} \geq \texttt{minimum\_demand\_from\_warehouse}[i], \quad \forall i \in [n_{\text{warehouses}}]. \tag{8}$$

Such formulations are incorrect because they do not enforce the logical dependency between the service allocation variables $x_{ij}$ and the warehouse opening decision $y_i$. The correct constraints should explicitly include the binary variable $y_i$ to ensure that demand can only be supplied when a warehouse is open. To verify executability and logical soundness, the solution evaluation agent instantiates the abstract parameterized model using a small synthetic dataset (a toy instance). Specifically, the agent generates dummy parameters (e.g., $\texttt{n\_warehouses = 10}, \texttt{n\_customers = 20}$), loads the instantiated model, and solves it using a standard solver. This process takes only milliseconds and allows the agent to inspect solver outputs and validate solution feasibility and consistency on the toy instance. By verifying correctness on a small but representative instantiation, Opt-Verifier ensures that the abstract formulation is executable and logically coherent without incurring the prohibitive cost of solving the full large-scale instance.

---

**Large-Scale Handling Process**

**Solution-Side Verification**
The solution-side verification process evaluates whether a candidate solution satisfies all modeled constraints and logical conditions. For a sampled toy instance, the solver returns the following warehouse opening decisions and total supplied demand:

- Warehouse 0 is open with a total supplied demand of 258.00.

- Warehouse 1 is open with a total supplied demand of 55.00.

- Warehouse 2 is open with a total supplied demand of 183.00.

- Warehouse 3 is open with a total supplied demand of 148.00.

- Warehouse 4 is open with a total supplied demand of 262.00.

- Warehouse 5 is not open. ...

---

**Large-Scale Handling Process**

The verification agent checks that all open warehouses satisfy their capacity limits and minimum demand requirements, while all closed warehouses supply zero demand. This process ensures both feasibility and logical consistency of the generated optimization model.

**Verification Comment**: The solution violates a core structural constraint. Warehouses 5 and 6 are marked as closed (non-operational), yet the optimal flow plan assigns 90 service units from Warehouse 5 to Customer 4, and allocates services from Warehouse 6 to Customers 9, 11, 13, and 19. This is a fundamental physical and mathematical contradiction, indicating a missing or incorrectly defined linking constraint that connects the facility activation variable to the service flow variable.

## C.4. Correctness Guarantee for Large-Scale Parameters

Opt-Verifier is an optimization modeling framework rather than a data entry system. We explicitly separate problem descriptions from parameter data to ensure correctness at scale. Large-scale coefficients (e.g., cost matrices or service allocation costs) are not manually typed by the LLM; instead, the LLM generates code to correctly load these parameters from external sources such as CSV files, databases, or APIs. The correctness of large-scale coefficients is therefore guaranteed by the data loading and binding process. If the code correctly reads and applies the parameters, the resulting optimization model is mathematically consistent with the intended large-scale instance.

## C.5. Additional Large-Scale Experimental Results

*Table 14.* Statistical information of the large-scale ComplexOR dataset.

| Metric | Variable Size | Constraint Size | Solving Time |
|---|---|---|---|
| Information | 18,759.4 | 33,568.0 | > 1000s |

*Table 15.* Modeling accuracy comparison across different methods.

| Method | Standard | CoT | CoE | OptiMUS | Ours |
|---|---|---|---|---|---|
| Modeling ACC (%) | 31.5 | 31.5 | 63.2 | 73.6 | **78.9** |

To further evaluate scalability, we construct large-scale instances based on the ComplexOR benchmark by replacing small-scale parameters with larger ones. The statistical properties of the resulting large-scale dataset are summarized in Table 14. Since directly solving these instances is computationally expensive (often exceeding 1,000 seconds), we rely on toy-parameter instantiations for solution-side verification and manually inspect the modeling logic for all methods.

The modeling accuracy results are reported in Table 15. Despite the increased scale, the modeling accuracy remains comparable to that on small-scale datasets, demonstrating that verification using toy parameters reliably reflects performance on large-scale problems. These results confirm that Opt-Verifier can effectively validate parameterized formulations and handle real-world large-scale optimization tasks.

Overall, by verifying abstract logic and validating executability on toy instances, Opt-Verifier ensures correctness and scalability for large-scale target problems without requiring direct solution of the full instance.

# D. Generalization Study

For the example to explain why Opt-Verifier can generalize on more general stochastic programming problems, we provide a more complex example to demonstrate the strong generalization ability.

**Problem** (simplified version) We study a grid-connected photovoltaic (PV) system with an energy storage system (ESS) over a 24-hour horizon, aiming to maximize the expected daily profit under uncertainties in PV generation, electricity prices, and local load demand. PV output is modeled as a uniform random variable, while electricity prices follow a mixed stochastic structure with deterministic peak prices and probabilistic off-peak and shoulder prices. The ESS is subject to capacity, state-of-charge, efficiency, and power constraints, and grid export is limited. The optimal operation schedule is determined to maximize expected profit while satisfying all operational constraints.

---

**Optimization Formulation**

**Optimization Model for PV–ESS Operation Given by Opt-Verifier:**

- **Decision Variables:**

  - $x_t^c$: ESS charging power at hour $t$ (kW).
  - $x_t^d$: ESS discharging power at hour $t$ (kW).
  - $x_t^{\text{grid}}$: Power exported to the grid at hour $t$ (kW).
  - $x_t^{\text{cur}}$: Curtailed PV power at hour $t$ (kW).
  - $x_t^{\text{shed}}$: Load shedding at hour $t$ (kW).
  - $E_t$: State of charge (SoC) of the ESS at the end of hour $t$ (kWh).

- **Objective Function:** Maximize the expected daily net profit over the 24-hour horizon:

$$\max \ \mathbb{E}\left[\sum_{t=1}^{24}\left(\lambda_t x_t^{\text{grid}} - C_{\text{cur}} x_t^{\text{cur}} - C_{\text{shed}} x_t^{\text{shed}}\right)\right].$$

- **Constraints:**

  1. *Power balance constraints:*

  $$P_t^{\text{PV}} + \eta_d x_t^d = D_t - x_t^{\text{shed}} + \frac{x_t^c}{\eta_c} + x_t^{\text{grid}} + x_t^{\text{cur}}, \quad \forall t \in T.$$

  2. *ESS state-of-charge dynamics:*

  $$E_t = E_{t-1} + x_t^c - \frac{x_t^d}{\eta_d \eta_c}, \quad \forall t \in T.$$

  3. *SoC limits and initial condition:*

  $$E_{\min} \leq E_t \leq E_{\max}, \quad \forall t \in T,$$

  $$E_0 = 300.$$

  4. *Non-simultaneous charging and discharging:*

  $$x_t^c \cdot x_t^d = 0, \quad \forall t \in T.$$

  5. *ESS power limits:*
  $$0 \leq x_t^c \leq P_{\max}^c, \quad 0 \leq x_t^d \leq P_{\max}^d, \quad \forall t \in T.$$

  6. *Grid export constraint:*
  $$0 \leq x_t^{\text{grid}} \leq P_{\max}^{\text{grid}}, \quad \forall t \in T.$$

  7. *Curtailment and load shedding non-negativity:*

  $$x_t^{\text{cur}} \geq 0, \quad x_t^{\text{shed}} \geq 0, \quad \forall t \in T.$$

# E. Case Study

For the example to explain why Opt-Verifier can mitigate the errors, we provide the following optimization problem with output of CoT and Opt-Verifier.

**Problem** (simplified version) In a vast network of canals interconnecting 9 strategic reservoirs, engineers have meticulously mapped out the maximum capacities for water transfer from reservoirs $i$ to $j$, i.e., $c_{ij}$. The challenge is to optimize this water distribution system to ascertain the maximum possible water transfer from Reservoir 0 to Reservoir 8 per day.

---

**Optimization Formulation**

**Optimization Model Given by CoT:**

- **Decision Variables:** $x_{ij}$: The amount of water transferred from Reservoir $i$ to Reservoir $j$, where $i, j = 0, 1, 2, \ldots, 8$.

- **Objective Function:** Maximize the water transfer from Reservoir 0 to 8:

$$Z = \sum_{j=1}^{8} x_{0j}$$

- **Constraints:**

  1. Water transfer capacities: $x_{ij} \leq c_{ij}$, for all $i, j$.
  2. Non-negativity: $x_{ij} \geq 0$, for all $i, j$.

---

This model is incorrect due to missing flow balance constraints. The verification process is outlined as follows:

- **Structure-Augmented Modeling:** The model references a maximum flow problem. It correctly formulates the flow balance constraint when recalling the standard model.

- **Structure-Side Verification:** The model interprets the current optimization model and compares it with the structure of the original problems.

- **Solution-Side Verification:** If the model lacks flow balance constraints, the obtained solution is represented as $x_{ij} = c_{ij}$. The evaluation agent in Opt-Verifier analyzes the solutions and determines that the inflow does not equal the outflow within the system. Consequently, the evaluation agent identifies this discrepancy as an error.

---

**Optimization Formulation**

**Optimization Model Given by Opt-Verifier:**

- **Modeling structures:**

    - **High Level:** Maximum flow problem
    - **Medium Level:** Single commodity maximum flow
    - **Low Level:**
        1. Directed Network: The flow is directed from one reservoir to another.
        2. Capacity Constraints: Each edge has a maximum capacity.
        3. Flow Conservation: The amount of water entering any intermediate reservoir must equal the amount leaving.

- **Decision Variables:** $x_{ij}$: The amount of water transferred from Reservoir $i$ to Reservoir $j$.

- **Objective Function:** Maximize the water transfer:

$$Z = \sum_{j=1}^{8} x_{0j}$$

- **Constraints:**

    1. $x_{ij} \leq c_{ij}$, for all $i, j$.
    2. $x_{ij} \geq 0$, for all $i, j$.
    3. Flow Conservation: $\sum_{\substack{j=0 \\ j \neq k}}^{8} x_{kj} = \sum_{\substack{i=0 \\ i \neq k}}^{8} x_{ik}$ for $k$ in the reservoirs

**Analysis**   The modeling structures are proposed to address the challenges of missing constraints. The core of structure-augmented modeling is to identify a similar standard optimization model, and identify the implicit constraints using the standard optimization model as a reference.

## F. Error Analysis on Different Problem Types

The results presented in Table 16 clearly demonstrate Opt-Verifier's strong generalization capabilities, as it consistently and significantly outperforms the CoT baseline across five distinct and challenging problem categories. This robust performance is particularly evident in problem types where standard prompting methods struggle. For instance, on the Capacitated TSP, where CoT achieves a mere 5.13% accuracy, Opt-Verifier boosts performance to 48.72%. Similarly, for Diet, Transportation, and Maximum Flow problems, Opt-Verifier elevates accuracy from the 16-27% range to a much more effective 55-82% range. This shows that Opt-Verifier's verification process can successfully navigate complex problem structures that are difficult for LLMs to model correctly. Furthermore, even in cases where the CoT baseline is already strong in some problems, such as the Facility Location-Allocation Problem (80.65%), Opt-Verifier still provides a significant improvement, pushing the accuracy to 93.55%. The consistent and substantial performance lift across this diverse set of problems underscores that Opt-Verifier's adaptive verification framework is a broadly applicable and effective strategy, rather than a technique tailored to a specific problem type.

*Table 16.* The performance on each problem category on the MAMO ComplexLP dataset

| Problem Category | CoT | Opt-Verifier |
|---|---|---|
| Diet Problem | 27.27% | 81.82% |
| Transportation Problem | 23.53% | 70.59% |
| Capacitated TSP | 5.13% | 48.72% |
| Maximum Flow Problem | 16.28% | 55.81% |
| Facility Location-Allocation Problem | 80.65% | 93.55% |

# G. The Prompt Design

## G.1. Structure Distillation

```
    interpretation_prompt=[
"""
You are a mathematical formulator working with a team of optimization experts. The
    ↪ objective is to tackle a complex optimization problem.
""",
"""
Please interpret and explain the following problem description.

{problem}

- What is the specific problem type of this OR and CO problem? What specific kind of OR
    ↪ problem?
""",
"""
This is the base formulation of the problem

{base_formulation}

- What is the subdivision of different kinds of this problem?
- Is this base formulation correct?
""",
"""
- Is there any implicit constraints in the problem, including but not limited to the
    ↪ logical selection relation, if/else and if/then relation?
""",
"""
Please summarize and write in JSON Format. For 'subdivision', please find the ones
    ↪ matching this problem description

```json
{{
  "problem_type": ..,
  "specific_type": ...,
  "subdivisions": {{
    subdivision 1: description,
    subdivision 2: description,
    ...
  }},
  "implicit_constraints": {{
    implicit constraint 1: description,
    implicit constraint 2: description,
    ...
  }},
}}
```

- Note that I'm going to use python json.loads() function to parse the json file, so
    ↪ please make sure the format is correct (don't add ',' before enclosing '}}' or ']'
    ↪ characters.
```

```
- Generate the complete json file and don't omit anything.
- Use '```json' and '```' to enclose the json file.
"""
]
```

## G.2. Structure-Augmented Modeling

```
formulation_prompt = [
"""
You are an expert mathematical formulator and an optimization professor at a top
    ↪ university. Your task is to model the problem in the standard LP or MILP form.
""",
"""
Here is the description of the problem to be formulated.

{problem}

- Please summarize the parameters and their tensor sizes.
- Please explain the definition of the parameters.
- Please keep the answer brief and concise.
""",
"""
please write in JSON Format. Make sure the bracket is closed, especially when processing
    ↪ the matrices. Do not transpose the matrices and keep the shape of the matrices.

{{
    "parameters": [
        {
            "symbol": "mathematical symbol of the parameters",
            "definition": "definition of the parameters","
            "value": the value of the parameters,
            "shape": [],
        },
        {
            "symbol": "mathematical symbol of the parameters",
            "definition": "definition of the parameters",
            "value": the value of the parameters,
            "shape": [],
        },
        ...
    ],
}}

- Use CamelCase and full words for new variable symbols, and do not include indices in the
    ↪  symbol (e.g. ItemsSold instead of itemsSold or items_sold or ItemsSold_i)
- Note that I'm going to use python json.loads() function to parse the json file, so
    ↪ please make sure the format is correct (don't add ',' before enclosing '}}' or ']'
    ↪ characters.
- Use '```json' and '```' to enclose the json file.
""",
"""
Here are some of the cases when we need auxiliary variables. Do we need to include
    ↪ auxiliary binary variables in the formulation?

- Logical Conditions: When a decision depends on a binary condition (e.g., whether to open
    ↪  a facility or not, use a kind of transportation or not ,and so on), auxiliary
    ↪ binary variables can represent these conditions.
- Modeling step costs: Using binary variables involves creating a mathematical formulation
    ↪  where costs change based on specific thresholds or levels of activity.
- Disjunctive Constraints: When a problem involves "either-or" situations, binary
    ↪ variables can be used to model these disjunctions effectively (Combined with the
    ↪ big M method).
- Capacity Constraints: In problems involving limited resources, binary variables can
    ↪ indicate whether a resource is being utilized or not, allowing for better modeling
    ↪ of capacity.
- Selection Problems: In scenarios where a fixed set of items or variables can be selected
    ↪  (e.g., choosing a subset of projects to fund), binary variables indicate the
    ↪ selection status.
- Scheduling Order: When determining the sequence in which tasks are performed, binary
    ↪ variables can indicate the order of tasks (e.g., Task A before Task B). This is
```

```
        ↪ often used in job-shop scheduling or project scheduling.
- Penalty Costs: In scheduling with penalties for delays (like tardiness or unmet
        ↪ deadlines), binary variables can help track whether a task incurs a penalty,
        ↪ allowing for cost minimization.
- Job Switching: In scenarios where workers or machines can switch between tasks, binary
        ↪ variables can indicate if a switch occurs, helping to manage transition times and
        ↪ costs.
""",
"""
This problem is a {problem_type} problem with structures

{structure}

To analyze the description carefully, here is the base formulation of this problem (which
        ↪ can be correct or needs to be modified)

{base_formulation}

Now take a deep breath and formulate this problem according to the description and base
        ↪ formulation.

- Consider whether we need to introduce auxiliary binary variables, note that do not
        ↪ include redundant variables.
- For variables, use integer type for discrete items (such as production, unit, people)
        ↪ and continuous ones for continuous items (water, land, time, grams, and so on).
- Your formulation should be in LaTeX mathematical format (do not include the $ symbols).
- Important: You can not define new parameters. You can only define new variables. Use
        ↪ CamelCase and full words for new variable symbols, and do not indices in the symbol
        ↪  (e.g. ItemsSold instead of itemsSold or items_sold or ItemsSold_i). You can
        ↪ include indices in the constraint and objective formulations.
- Make sure that you do not use the numeric number in the formulation except when
        ↪ necessary, instead, you use the parameter name (you can include indices in the
        ↪ constraint and objective formulations).
- Always use non-strict inequalities (e.g. \\leq instead of <), even if the constraint is
        ↪ strict.

Take a deep breath and solve the problem step by step.
"""
]
```

### G.3. Structure Interpretation and Structure Consistency Verification

```
modification_prompt = [
"""
You are an expert mathematical formulator and an optimization professor at a top
    ↪ university. Your task is to model and fix the problem in the standard LP or MILP
    ↪ form.
""",
"""
This is a {problem_type} problem with parameters

{parameters}

The formulation is as follows

{formulation_interpretation}

Does this problem consistent with the characteristics of the following structure
    ↪ description? If yes, please say "Yes" directly.
If not, please give your comments to modify the formulation.

{original_problem_interpretation}
""",
"""
Please reformulate the problem to make the formulation consistent with the structure
    ↪ description.

- Consider whether we need to introduce extra binary variables or linearization for a
    ↪ piece-wise linear function.
- Your formulation should be in LaTeX mathematical format (do not include the $ symbols).
- Important: You can not define new parameters. You can only define new variables. Use
    ↪ CamelCase and full words for new variable symbols, and do not include indices in
    ↪ the symbol (e.g. ItemsSold instead of itemsSold or items_sold or ItemsSold_i). You
    ↪ can include indices in the constraint and objective formulations.
- Make sure that you do not use a numeric number in the formulation except where necessary
    ↪ ; instead, you use the parameter name (you can include indices in the constraint
    ↪ and objective formulations).
- Always use non-strict inequalities (e.g. \\leq instead of <), even if the constraint is
    ↪ strict.

Take a deep breath and solve the problem step by step.
"""
]
```

## G.4. Solution Interpretation and Solution Validity Verification

```
solution_prompt = [
"""
You are an expert mathematical formulator and an optimization professor at a top
    ↪ university. Your task is to model and fix the problem using the solution
    ↪ information in the standard LP or MILP form.
""",
"""
This is a {problem_type} problem with solutions

{solutions}

The formulation is as follows

{formulation_interpretation}

Please interpret the meaning of the solution.
""",
"""
Here is the problem description.

{original_problem_interpretation}

Is this solution the optimal solution? The optimal solution should be mathematical sound
    ↪ and logical coherence:
- We cannot find a better solution.
- The solution should meet the constraints of the problem description.

If yes, please say "Yes" directly.
If not, please give your comments to modify the formulation.
""",
"""
Please reformulate the problem to make the formulation consistent with the structure
    ↪ description.

- Consider whether we need to introduce extra binary variables or linearization for a
    ↪ piece-wise linear function.
- Your formulation should be in LaTeX mathematical format (do not include the $ symbols).
- Important: You can not define new parameters. You can only define new variables. Use
    ↪ CamelCase and full words for new variable symbols, and do not include indices in
    ↪ the symbol (e.g. ItemsSold instead of itemsSold or items_sold or ItemsSold_i). You
    ↪ can include indices in the constraint and objective formulations.
- Make sure that you do not use a numeric number in the formulation except where necessary
    ↪ ; instead, you use the parameter name (you can include indices in the constraint
    ↪ and objective formulations).
- Always use non-strict inequalities (e.g. \\leq instead of <), even if the constraint is
    ↪ strict.

Take a deep breath and solve the problem step by step.
"""
]
```

# H. More Examples

## H.1. NL4Opt

---

**Example in NL4Opt**

**Natural Language Description:**
A fishery wants to transport their catch. They can either use local sled dogs or trucks. Local sled dogs can take 100 fish per trip while trucks can take 300 fish per trip. The cost per trip for sled dogs is $50 while the cost per trip for a truck is $100. The budget is at most $1000 and the number of sled dog trips must be less than the number of truck trips. Formulate an LP to maximize the number of fish that can be transported.

- - - - - - - - - - - - - - - - - - - - - - - - - - - - - - - - - - - - - - - - -

**Modeling Structure Given by Opt-Verifier:**

```
1  {
2    "problem_type": "Linear Programming",
3    "specific_type": "Resource Allocation Problem",
4    "subdivisions": {
5      "transportation optimization": "Optimizing the transport of goods or
           ↪ resources using various modes of transport.",
6      "resource allocation": "Distributing limited resources to achieve the
           ↪ best possible outcome.",
7      "cost minimization and profit maximization": "Balancing costs against
           ↪ outputs to maximize profit or benefits."
8    },
9    "implicit_constraints": {
10     "non-negative trips": "The number of trips (x and y) cannot be negative."
           ↪ ,
11     "integer trips": "The number of trips should generally be an integer
           ↪ value."
12   }
13 }
```

- - - - - - - - - - - - - - - - - - - - - - - - - - - - - - - - - - - - - - - - -

**Optimization Model Given by Opt-Verifier:**

```
1  {
2      "constraints": [
3          {
4              "name": "Budget Constraint",
5              "description": "The total cost of trips must not exceed the budget
                   ↪ .",
6              "formulation": "CostPerSledDogTrip \\cdot NumberOfSledDogTrips +
                   ↪ CostPerTruckTrip \\cdot NumberOfTruckTrips \\leq TotalBudget"
7          },
8          {
9              "name": "Trip Comparison Constraint",
10             "description": "The number of sled dog trips must not exceed the
                   ↪ number of truck trips.",
11             "formulation": "NumberOfSledDogTrips \\leq NumberOfTruckTrips"
12         },
13         {
14             "name": "Non-negativity Constraints",
```

```
15        "description": "The number of trips must be non-negative.",
16        "formulation": "NumberOfSledDogTrips \\geq 0"
17    },
18    {
19        "name": "Non-negativity Constraints",
20        "description": "The number of trips must be non-negative.",
21        "formulation": "NumberOfTruckTrips \\geq 0"
22    },
23    {
24        "name": "Integer Constraints for Sled Dog Trips",
25        "description": "The number of sled dog trips must be an integer.",
26        "formulation": "NumberOfSledDogTrips \\in \\mathbb{Z}"
27    },
28    {
29        "name": "Integer Constraints for Truck Trips",
30        "description": "The number of truck trips must be an integer.",
31        "formulation": "NumberOfTruckTrips \\in \\mathbb{Z}"
32    }
33  ],
34  "variables": [
35    {
36        "definition": "Integer variable representing the number of sled dog
            ↪  trips.",
37        "symbol": "NumberOfSledDogTrips",
38        "type": "integer",
39        "shape": []
40    },
41    {
42        "definition": "Integer variable representing the number of truck
            ↪ trips.",
43        "symbol": "NumberOfTruckTrips",
44        "type": "integer",
45        "shape": []
46    }
47  ],
48  "objective": [
49    {
50        "description": "Maximize the total number of fish transported.",
51        "formulation": "Z = FishPerSledDogTrip \\cdot NumberOfSledDogTrips
            ↪ + FishPerTruckTrip \\cdot NumberOfTruckTrips",
52        "objective_sense": "Maximize"
53    }
54  ]
55 }
```

**Solver Code Given by Opt-Verifier:**

```
import json
import numpy as np
import math
import gurobipy as gp
from gurobipy import GRB
```

```python
# Create a new model
model = gp.Model('model')

# Define parameters
parameters = {
    "CostPerSledDogTrip": 50,
    "CostPerTruckTrip": 100,
    "FishPerSledDogTrip": 100,
    "FishPerTruckTrip": 300,
    "TotalBudget": 1000
}

# Define variables
NumberOfSledDogTrips = model.addVar(vtype=GRB.INTEGER, name="NumberOfSledDogTrips")
NumberOfTruckTrips = model.addVar(vtype=GRB.INTEGER, name="NumberOfTruckTrips")

# Add variables to the model
model.update()

# Constraints
# Budget Constraint
model.addConstr(
    parameters["CostPerSledDogTrip"] * NumberOfSledDogTrips +
    parameters["CostPerTruckTrip"] * NumberOfTruckTrips <=
    parameters["TotalBudget"],
    name="BudgetConstraint"
)

# Trip Comparison Constraint
model.addConstr(
    NumberOfSledDogTrips <= NumberOfTruckTrips,
    name="TripComparisonConstraint"
)

# Non-negativity Constraints
model.addConstr(NumberOfSledDogTrips >= 0, name="NonNegativitySledDogTrips")
model.addConstr(NumberOfTruckTrips >= 0, name="NonNegativityTruckTrips")

# Set the objective
model.setObjective(
    parameters["FishPerSledDogTrip"] * NumberOfSledDogTrips +
    parameters["FishPerTruckTrip"] * NumberOfTruckTrips,
    GRB.MAXIMIZE
)

# Optimize the model
model.optimize()
status = model.status

obj_val = None
# Check whether the model is infeasible, has infinite solutions, or has an optimal
    ↪ solution
if status == gp.GRB.INFEASIBLE:
    obj_val = "infeasible"
elif status == gp.GRB.UNBOUNDED:
    obj_val = "unbounded"
elif status == gp.GRB.OPTIMAL:
    obj_val = model.objVal

print("Objective Value:", obj_val)
```

## H.2. Mamo ComplexLP

### Example in Mamo ComplexLP

**Natural Language Description:**
In a scenario involving a salesperson who needs to visit five different cities to conduct business, each city is uniquely numbered from 1 to 5. The salesperson's objective is to minimize the total travel expenses, which could be influenced by factors such as distance, fuel costs, or transportation fees. The salesperson can start their journey from any of these cities but must ensure they visit each city exactly once before returning to their starting point.

The travel costs between the cities are as follows:
- From City 1, the travel costs are 58 units to City 2, 15 units to City 3, 75 units to City 4, and 91 units to City 5.
- From City 2, it costs 58 units to City 1, 54 units to City 3, 85 units to City 4, and 11 units to City 5.
- Traveling from City 3, the expenses are 15 units to City 1, 54 units to City 2, 28 units to City 4, and 61 units to City 5.
- From City 4, the costs are 75 units to City 1, 85 units to City 2, 28 units to City 3, and 47 units to City 5.
- Lastly, from City 5, it costs 91 units to City 1, 11 units to City 2, 61 units to City 3, and 47 units to City 4.

Given this setup, what is the minimum total travel cost for the salesperson to visit each city exactly once and then return to the starting city?

- - - - - - - - - - - - - - - - - - - - - - - - - - - - - - - - - - - - - - - - - - - - - -

**Modeling Structure Given by Opt-Verifier:**

```
1  {
2    "problem_type": "Combinatorial Optimization Problem",
3    "specific_type": "Traveling Salesman Problem (TSP)",
4    "subdivisions": {
5      "Hamiltonian cycle": "Path that visits each vertex exactly once and
         ↪ returns to the starting vertex",
6      "Weighted graph": "Graph with weights on edges representing travel costs
         ↪ between cities",
7      "Directed graph": "Graph where edges have a direction, indicating the
         ↪ cost of travel from one city to another"
8    },
9    "implicit_constraints": {
10     "subtour elimination": "Explicit constraints to prevent subtours in the
         ↪ solution",
11     "start_end city constraint": "Salesperson must start and end at the same
         ↪ city"
12   }
13 }
```

- - - - - - - - - - - - - - - - - - - - - - - - - - - - - - - - - - - - - - - - - - - - - -

**Optimization Model Given by Opt-Verifier:**

```
1  {
2      "constraints": [
3          {
4              "name": "Each City Visited Once",
5              "description": "Each city must be visited exactly once by the
                 ↪ salesperson.",
6              "formulation": "\\sum_{j \\in Cities} x_{ij} = 1\\quad \\forall i
                 ↪ \\in Cities"
```

```
 7          },
 8          {
 9              "name": "Return to Start City",
10              "description": "The salesperson must return to the starting city
                    ↪ after visiting all cities.",
11              "formulation": "\\sum_{i \\in Cities} x_{ji} = 1\\quad \\forall j
                    ↪ \\in Cities"
12          },
13          {
14              "name": "Subtour Elimination",
15              "description": "Constraints to prevent subtours in the solution.",
16              "formulation": "u_i - u_j + (|Cities|) \\cdot x_{ij} \\leq |Cities|
                    ↪ - 1\\quad \\forall i, j \\in Cities, \\; i \\neq j"
17          },
18          {
19              "name": "Position Constraints",
20              "description": "Position variables must be within valid range.",
21              "formulation": "2 \\leq u_i \\leq |Cities| \\quad \\forall i \\in
                    ↪ Cities"
22          }
23      ],
24      "variables": [
25          {
26              "definition": "Binary variable indicating whether the salesperson
                    ↪ travels from city i to city j.",
27              "symbol": "x_ij",
28              "type": "binary",
29              "shape": [
30                  5,
31                  5
32              ]
33          },
34          {
35              "definition": "Auxiliary continuous variable representing the
                    ↪ position of city i in the tour.",
36              "symbol": "u_i",
37              "type": "continuous",
38              "shape": [
39                  5
40              ]
41          }
42      ],
43      "objective": [
44          {
45              "description": "Minimize the total travel cost for the salesperson
                    ↪ .",
46              "formulation": "Z = \\sum_{i \\in Cities} \\sum_{j \\in Cities} d_{
                    ↪ ij} \\cdot x_{ij}",
47              "objective_sense": "Minimize"
48          }
49      ]
50  }
```

------------------------------------------------

**Solver Code Given by Opt-Verifier:**

```python
import json
import numpy as np
import math
import gurobipy as gp
from gurobipy import GRB

# Create a new model
model = gp.Model('model')

# Define parameters
parameters = {
    "Cities": [1, 2, 3, 4, 5],
    "TravelCosts": [
        [0, 58, 15, 75, 91],
        [58, 0, 54, 85, 11],
        [15, 54, 0, 28, 61],
        [75, 85, 28, 0, 47],
        [91, 11, 61, 47, 0]
    ]
}

Cities = parameters["Cities"]
TravelCosts = parameters["TravelCosts"]
num_cities = len(Cities) # Number of cities

# Define variables
x = model.addVars(num_cities, num_cities, vtype=GRB.BINARY, name="x") # Binary
    ↪ variables for routes
u = model.addVars(num_cities, vtype=GRB.CONTINUOUS, name="u") # Continuous variables
    ↪  for subtour elimination

# Constraints: Each City Visited Once
for i in range(num_cities):
    model.addConstr(gp.quicksum(x[i, j] for j in range(num_cities) if i != j) == 1, f
        ↪ "VisitOnce_Cities[{i}]")

# Constraints: Return to Start City
for j in range(num_cities):
    model.addConstr(gp.quicksum(x[i, j] for i in range(num_cities) if i != j) == 1, f
        ↪ "ReturnToStart_Cities[{j}]")

# Subtour Elimination Constraints
for i in range(1, num_cities):
    for j in range(1, num_cities):
        if i != j:
            model.addConstr(u[i] - u[j] + num_cities * x[i, j] <= num_cities - 1, f"
                ↪ SubtourElimination_Cities[{i}][{j}]")

# Position Constraints
for i in range(num_cities):
    model.addConstr(u[i] >= 2, f"LowerBound_u[{i}]")
    model.addConstr(u[i] <= num_cities, f"UpperBound_u[{i}]")

# Objective: Minimize total travel cost
model.setObjective(gp.quicksum(TravelCosts[i][j] * x[i, j] for i in range(num_cities
    ↪ ) for j in range(num_cities)), GRB.MINIMIZE)
```

```python
# Optimize the model
model.optimize()
status = model.status

obj_val = None
# Check whether the model is infeasible, has infinite solutions, or has an optimal
    ↪ solution
if status == gp.GRB.INFEASIBLE:
    obj_val = "infeasible"
elif status == gp.GRB.UNBOUNDED:
    obj_val = "unbounded"
elif status == gp.GRB.OPTIMAL:
    obj_val = model.objVal

print("Objective Value:", obj_val)
```

## H.3. ComplexOR

---

**Example in ComplexOR**

**Natural Language Description:**

The capacitated warehouse location problem involves determining the optimal locations for a set number of warehouses to service customers at minimum cost, taking into account warehouse capacities, operating costs, and customer demand.

The capacitated warehouse location problem is the problem of locating NumberOfLocations warehouses which have to service NumberOfCustomers customers, at minimum cost. Each customer has an associated demand CustomerDemand. There are constraints on the total demand that can be met from a warehouse, as specified by WarehouseCapacity. Costs are incurred when allocating service to customers from warehouses ServiceAllocationCost, and warehouses have a fixed operating cost WarehouseFixedCost. Additionally, there is a lower limit MinimumDemandFromWarehouse on the amount of demand that a warehouse must meet if it is opened, as well as constraints on the minimum MinimumOpenWarehouses and maximum MaximumOpenWarehouses number of warehouses that can be operational.

The total number of potential warehouse locations is 10. The total number of customers to be serviced is 20. The demand of each customer is [117, 86, 69, 53, 110, 74, 136, 140, 126, 79, 54, 86, 114, 76, 136, 73, 144, 51, 53, 120]. The cost of allocating service from each warehouse to each customer is [[80, 94, 44, 51, 190, 44, 129, 178, 129, 91, 172, 119, 177, 150, 90, 51, 53, 97, 184, 87], [139, 33, 104, 135, 50, 176, 97, 121, 47, 29, 186, 163, 149, 108, 156, 169, 100, 160, 153, 85], [153, 36, 18, 170, 18, 181, 178, 68, 171, 106, 159, 110, 21, 106, 91, 29, 144, 140, 155, 116], [103, 59, 78, 125, 14, 11, 152, 95, 76, 173, 36, 148, 75, 132, 59, 153, 113, 74, 185, 71], [193, 186, 130, 145, 114, 150, 33, 154, 20, 75, 103, 30, 137, 131, 167, 32, 53, 150, 176, 166], [159, 130, 156, 65, 36, 59, 199, 124, 104, 72, 180, 73, 43, 152, 143, 90, 161, 65, 172, 141], [173, 121, 110, 127, 22, 159, 195, 137, 47, 10, 87, 11, 154, 66, 126, 60, 152, 54, 20, 25], [181, 34, 186, 152, 109, 195, 133, 198, 30, 65, 69, 19, 109, 143, 108, 196, 59, 133, 10, 123], [82, 113, 147, 21, 88, 24, 38, 16, 70, 122, 148, 192, 116, 108, 18, 20, 143, 18, 116, 142], [176, 170, 87, 91, 195, 183, 124, 89, 72, 97, 89, 23, 45, 196, 97, 27, 83, 81, 171, 148]]. The total capacity for each warehouse is [3010, 2910, 4530, 4720, 4920, 3750, 4930, 2970, 3310, 2460]. The lower limit on the demand that must be met from a warehouse if it is to be operational is [64, 55, 27, 71, 93, 90, 89, 87, 43, 50]. The minimum number of warehouses that need to be operational is 3. The maximum number of warehouses that can be operational is 8. The fixed operating cost of each warehouse is [8517, 5068, 9433, 6127, 6033, 5966, 7762, 9406, 6602, 7040].

- - - - - - - - - - - - - - - - - - - - - - - - - - - - - - - - - - - - - - - - - - - - - - - - - - - -

**Modeling Structure Given by Opt-Verifier:**

```
1  {
2    "problem_type": "Mixed Integer Linear Programming",
3    "specific_type": "Capacitated Warehouse Location Problem",
4    "subdivisions": {
5      "1": "Location Optimization",
6      "2": "Capacity Planning",
7      "3": "Cost Minimization"
8    },
9    "implicit_constraints": {
10     "1": "Each customer's demand must be fully met.",
11     "2": "A warehouse that is opened must meet a specified minimum demand."
12   }
13 }
```

- - - - - - - - - - - - - - - - - - - - - - - - - - - - - - - - - - - - - - - - - - - - - - - - - - - -

**Optimization Model Given by Opt-Verifier:**

```
1  {
2      "constraints": [
```

```
3          {
4              "name": "Demand Meeting",
5              "description": "Each customer's demand must be fully met.",
6              "formulation": "\\sum_{j=1}^{NumberOfLocations} y_{ij} =
                   ↪ CustomerDemand[i] \\quad \\forall i"
7          },
8          {
9              "name": "Capacity Limit",
10             "description": "The total demand served from each warehouse cannot
                   ↪ exceed its capacity.",
11             "formulation": "\\sum_{i=1}^{NumberOfCustomers} y_{ij} \\leq
                   ↪ WarehouseCapacity[j] \\cdot x_j \\quad \\forall j"
12         },
13         {
14             "name": "Minimum Demand",
15             "description": "A warehouse that is opened must meet a specified
                   ↪ minimum demand.",
16             "formulation": "y_{ij} \\geq MinimumDemandFromWarehouse[j] \\cdot
                   ↪ x_j \\quad \\forall j, \\forall i"
17         },
18         {
19             "name": "Minimum Open Warehouses",
20             "description": "At least a minimum number of warehouses must be
                   ↪ operational.",
21             "formulation": "\\sum_{j=1}^{NumberOfLocations} x_j \\geq
                   ↪ MinimumOpenWarehouses"
22         },
23         {
24             "name": "Maximum Open Warehouses",
25             "description": "At most a maximum number of warehouses can be
                   ↪ operational.",
26             "formulation": "\\sum_{j=1}^{NumberOfLocations} x_j \\leq
                   ↪ MaximumOpenWarehouses"
27         }
28     ],
29     "variables": [
30         {
31             "definition": "Binary variable indicating whether warehouse j is
                   ↪ opened (1) or not (0)",
32             "symbol": "x_j",
33             "type": "binary",
34             "shape": [
35                 10
36             ]
37         },
38         {
39             "definition": "Continuous variable representing the amount of
                   ↪ demand served from warehouse j to customer i",
40             "symbol": "y_{ij}",
41             "type": "continuous",
42             "shape": [
43                 20,
```

```
44              10
45          ]
46      }
47   ],
48   "objective": [
49      {
50          "description": "Minimize the total cost of servicing customers,
              ↪ including service allocation and operating costs of
              ↪ warehouses.",
51          "formulation": "Z = \\sum_{j=1}^{NumberOfLocations}
              ↪ WarehouseFixedCost[j] \\cdot x_j + \\sum_{i=1}^{
              ↪ NumberOfCustomers} \\sum_{j=1}^{NumberOfLocations}
              ↪ ServiceAllocationCost[j][i] \\cdot y_{ij}",
52          "objective_sense": "Minimize"
53      }
54   ]
55 }
```

---

**Solver Code Given by Opt-Verifier:**

```python
import json
import numpy as np
import math
import gurobipy as gp
from gurobipy import quicksum # Import quicksum

# Create a Gurobi model
model = gp.Model('model')

# Parameters
NumberOfLocations = 10
NumberOfCustomers = 20

CustomerDemand = [117, 86, 69, 53, 110, 74, 136, 140, 126, 79,
            54, 86, 114, 76, 136, 73, 144, 51, 53, 120]

ServiceAllocationCost = [
   [80, 94, 44, 51, 190, 44, 129, 178, 129, 91, 172, 119, 177, 150, 90, 51, 53, 97,
       ↪ 184, 87],
   [139, 33, 104, 135, 50, 176, 97, 121, 47, 29, 186, 163, 149, 108, 156, 169, 100,
       ↪ 160, 153, 85],
   [153, 36, 18, 170, 18, 181, 178, 68, 171, 106, 159, 110, 21, 106, 91, 29, 144,
       ↪ 140, 155, 116],
   [103, 59, 78, 125, 14, 11, 152, 95, 76, 173, 36, 148, 75, 132, 59, 153, 113, 74,
       ↪ 185, 71],
   [193, 186, 130, 145, 114, 150, 33, 154, 20, 75, 103, 30, 137, 131, 167, 32, 53,
       ↪ 150, 176, 166],
   [159, 130, 156, 65, 36, 59, 199, 124, 104, 72, 180, 73, 43, 152, 143, 90, 161, 65,
       ↪ 172, 141],
   [173, 121, 110, 127, 22, 159, 195, 137, 47, 10, 87, 11, 154, 66, 126, 60, 152, 54,
       ↪ 20, 25],
   [181, 34, 186, 152, 109, 195, 133, 198, 30, 65, 69, 19, 109, 143, 108, 196, 59,
       ↪ 133, 10, 123],
   [82, 113, 147, 21, 88, 24, 38, 16, 70, 122, 148, 192, 116, 108, 18, 20, 143, 18,
       ↪ 116, 142],
```

```
    [176, 170, 87, 91, 195, 183, 124, 89, 72, 97, 89, 23, 45, 196, 97, 27, 83, 81,
        ↪ 171, 148]
]

WarehouseCapacity = [3010, 2910, 4530, 4720, 4920, 3750, 4930, 2970, 3310, 2460]

WarehouseFixedCost = [8517, 5068, 9433, 6127, 6033, 5966, 7762, 9406, 6602, 7040]

MinimumDemandFromWarehouse = [64, 55, 27, 71, 93, 90, 89, 87, 43, 50]

MinimumOpenWarehouses = 3
MaximumOpenWarehouses = 8

# Variables
x = model.addVars(NumberOfLocations, vtype=gp.GRB.BINARY, name="x")
y = model.addVars(NumberOfCustomers, NumberOfLocations, vtype=gp.GRB.CONTINUOUS,
    ↪ name="y")

# Objective function
model.setObjective(
    quicksum(WarehouseFixedCost[j] * x[j] for j in range(NumberOfLocations)) +
    quicksum(quicksum(ServiceAllocationCost[j][i] * y[i, j] for j in range(
        ↪ NumberOfLocations)) for i in range(NumberOfCustomers)),
    gp.GRB.MINIMIZE
)

# Constraints
for i in range(NumberOfCustomers):
    model.addConstr(
        quicksum(y[i, j] for j in range(NumberOfLocations)) == CustomerDemand[i],
        name=f"demand_meeting_{i}"
    )

for j in range(NumberOfLocations):
    model.addConstr(
        quicksum(y[i, j] for i in range(NumberOfCustomers)) <= WarehouseCapacity[j] *
            ↪ x[j],
        name=f"capacity_limit_{j}"
    )

for j in range(NumberOfLocations):
    model.addConstr(
        quicksum(y[i, j] for i in range(NumberOfCustomers)) >=
            ↪ MinimumDemandFromWarehouse[j] * x[j],
        name=f"minimum_demand_{j}"
    )

model.addConstr(
    quicksum(x[j] for j in range(NumberOfLocations)) >= MinimumOpenWarehouses,
    name="minimum_open_warehouses"
)

model.addConstr(
    quicksum(x[j] for j in range(NumberOfLocations)) <= MaximumOpenWarehouses,
    name="maximum_open_warehouses"
)

# Optimize the model
model.optimize()

# Check the optimization status
status = model.status
```

```python
obj_val = None
if status == gp.GRB.INFEASIBLE:
    obj_val = "infeasible"
elif status == gp.GRB.UNBOUNDED:
    obj_val = "unbounded"
elif status == gp.GRB.OPTIMAL:
    obj_val = model.objVal

print("Objective Value:", obj_val)
```

## H.4. ComplexOR

> **Example in ComplexOR**
>
> **Natural Language Description:**
> The Knapsack Problem involves selecting the most valuable combination of items to fit in a knapsack without exceeding its weight limit. The Knapsack Problem is a classic optimization problem in operations research and computer science. The problem is to determine the most valuable combination of items to include in a knapsack, given a set of TotalItems with different values and weights represented by ItemValues and ItemWeights respectively, and a maximum weight capacity of the knapsack MaxKnapsackWeight. The goal is to maximize the total value of the items in the knapsack, represented by ItemValues, without exceeding its weight capacity MaxKnapsackWeight. The available kinds of items is 6. The value of each kind of item is [17, 4, 10, 21, 12, 18]. The weight of each item is [23, 6, 14, 30, 15, 25]. The maximum weight capacity of the knapsack is 60. The total weight of the selected items must not exceed MaxKnapsackWeight. Multiple items in one kind can be selected. No more than TotalItems kinds of items can be considered for selection. Maximize the total value of the items in the knapsack.
>
> ---
>
> **Modeling Structure Given by Opt-Verifier:**
>
> ```
> 1  {
> 2    "problem_type": "Combinatorial Optimization",
> 3    "specific_type": "Bounded Knapsack Problem",
> 4    "subdivisions": {
> 5      "Subdivision 1": "0/1 Knapsack Problem",
> 6      "Subdivision 2": "Bounded Knapsack Problem",
> 7      "Subdivision 3": "Multi-dimensional Knapsack Problem"
> 8    },
> 9    "implicit_constraints": {
> 10     "implicit constraint 1": "x_i must be an integer",
> 11     "implicit constraint 2": "MaxKnapsackWeight must be non-negative"
> 12   }
> 13 }
> ```
>
> ---
>
> **Optimization Model Given by Opt-Verifier:**
>
> ```
> 1  {
> 2    "constraints": [
> 3      {
> 4        "name": "Weight Limit Constraint",
> 5        "description": "The total weight of selected items must not exceed
>              ↪ the maximum weight capacity of the knapsack.",
> 6        "formulation": "\\sum_{i=1}^{TotalItems} ItemWeights_i \\cdot
>              ↪ ItemQuantities_i \\leq MaxKnapsackWeight"
> 7      },
> 8      {
> 9        "name": "Non-negativity and Integer Constraints",
> 10       "description": "The quantity of each item selected must be non-
>              ↪ negative and integer.",
> 11       "formulation": "ItemQuantities_i \\geq 0\\quad \\text{and integer
>              ↪ for }i = 1,2,\\ldots, TotalItems"
> 12     }
> 13   ],
> 14   "variables": [
> ```

```
15          {
16              "definition": "Number of items of type i selected (i = 1to
                    ↪ TotalItems)",
17              "symbol": "ItemQuantities_i",
18              "type": "integer",
19              "shape": []
20          }
21      ],
22      "objective": [
23          {
24              "description": "Maximize the total value of the selected items in
                    ↪ the knapsack.",
25              "formulation": "Z = \\sum_{i=1}^{TotalItems} ItemValues_i \\cdot
                    ↪ ItemQuantities_i",
26              "objective_sense": "Maximize"
27          }
28      ]
29  }
```

**Solver Code Given by Opt-Verifier:**

```python
import json
import numpy as np
import math
import gurobipy as gp

# Create a new model
model = gp.Model('model')

# Parameters
TotalItems = 6
ItemValues = [17, 4, 10, 21, 12, 18]
ItemWeights = [23, 6, 14, 30, 15, 25]
MaxKnapsackWeight = 60

# Variables: ItemQuantities_i (integer variables)
ItemQuantities = model.addVars(TotalItems, vtype=gp.GRB.INTEGER, name="
    ↪ ItemQuantities")

# Objective: Maximize Z = sum(ItemValues_i * ItemQuantities_i)
model.setObjective(gp.quicksum(ItemValues[i] * ItemQuantities[i] for i in range(
    ↪ TotalItems)), gp.GRB.MAXIMIZE)

# Constraints
# Weight Limit Constraint: sum(ItemWeights_i * ItemQuantities_i) <=
    ↪ MaxKnapsackWeight
model.addConstr(gp.quicksum(ItemWeights[i] * ItemQuantities[i] for i in range(
    ↪ TotalItems)) <= MaxKnapsackWeight, "WeightLimit")

# Non-negativity and Integer Constraints are inherently defined by the variable type
# ItemQuantities_i >= 0 and ItemQuantities_i in Z
# Gurobi automatically treats integer variables as non-negative, so no additional
    ↪ constraint is needed for non-negativity.

# Optimize the model
```

```python
model.optimize()
status = model.status

obj_val = None
# Check whether the model is infeasible, has infinite solutions, or has an optimal
    ↪ solution
if status == gp.GRB.INFEASIBLE:
    obj_val = "infeasible"
elif status == gp.GRB.UNBOUNDED:
    obj_val = "unbounded"
elif status == gp.GRB.OPTIMAL:
    obj_val = model.objVal

print("Objective Value:", obj_val)
```

