# OpenReview forum: "Opt-Verifier: Unleashing the Power of LLMs for Optimization Modeling via Dual-Side Verification"
_ICML.cc/2026/Conference — ICML 2026 regular_

### Official Review · Reviewer_TC8b · 2026-02-17

**Soundness:** 3
**Presentation:** 4
**Significance:** 3
**Originality:** 3
**Overall Recommendation:** 4
**Confidence:** 4

**Summary:**

Large Language Models have shown promise in automating operations research modeling, yet they frequently generate optimization models that are executable but fundamentally flawed, producing logical contradictions that code compilers cannot detect . To overcome these limitations, the authors introduce Opt-Verifier, a multi-agent framework that implements a novel dual-side verification mechanism to ensure both structural and solution correctness. The process begins with structure-augmented modeling, where an agent distills a multi-level problem structure (from high-level problem types to low-level constraints) to guide the initial formulation. This is followed by structure-side verification, where a "back-translation" agent converts the generated mathematical model back into an abstract description to detect discrepancies against the original problem requirements.

The framework then employs solution-side verification to catch logical errors that do not trigger runtime exceptions . In this step, the generated model is solved using a "toy instantiation" of the data, and the resulting numerical values are interpreted into natural language (e.g., verifying if flow conservation holds) for an LLM evaluator to critique . Extensive experiments across five benchmarks—NL4Opt, Mamo ComplexLP, ComplexOR, IndustryOR, and OptMATH—demonstrate that Opt-Verifier significantly outperforms existing methods, including chain-of-thought baselines and fine-tuned models like ORLM. It achieves an average accuracy improvement of approximately 20% over standard prompting methods and using Opt-Verifier on GPT-4o-mini surpasses frontier-level reasoning models such as DeepSeek-R1 and o1 on several complex tasks.

**Compliance With Llm Reviewing Policy:**

Affirmed.

**Final Justification:**

I think the authors have effectively addressed my remaining concerns and I am willing to raise my score.

**Key Questions For Authors:**

**Major questions**

*I would consider raising the score if the author could address my following questions and / or the weakness points:*

- How does the performance of Opt-Verifier compare to simply using the base model with standard test-time scaling approaches, such as majority voting or multi-round self-correction without the dual-side framework? It would be interesting to know if Opt-Verifier is more token-efficient or effective than these simpler, established baselines.

- How do you ensure that the "teacher" or "verifier" models in the dual-side verification steps are correct? Since they are based on the same underlying LLM as the generator, is there a risk that they share the same biases or make the same interpretation errors, leading to a confirmation bias where incorrect models are verified as correct?

**Minor questions**

- Page 7, Table 2 Caption: There appears to be a typo in the caption for Table 2.

- Appendix G Formatting: The syntax highlighting for the prompts in Appendix G looks a bit fuzzy, particularly the coloring of the JSON structure. It might be helpful to check the language setting used for the code blocks.

**Limitations:**

Several recent works in the NL2OPT space, such as LLMOPT [1], OptiMind [2], and AlphaOpt [3], also utilize iterative self-correction or structure-augmented modeling. Could you clarify the specific methodological differences between Opt-Verifier and these approaches?
I recommend citing these works and, if possible, benchmarking against them to contextualize the contribution.

[1] Jiang, Caigao, et al. "LLMOPT: Learning to define and solve general optimization problems from scratch." arXiv preprint arXiv:2410.13213 (2024).

[2] Zhang, Xinzhi, et al. "OptiMind: Teaching LLMs to think like optimization experts." arXiv preprint arXiv:2509.22979 (2025).

[3] Kong, Minwei, et al. "AlphaOPT: Formulating Optimization Programs with Self-Improving LLM Experience Library." arXiv preprint arXiv:2510.18428 (2025).

**Strengths And Weaknesses:**

**Strengths**

- The proposed "Dual-Side Verification" framework is a novel contribution to the field of automated optimization modeling.

- The experimental validation is robust and extensive. The authors evaluate the method against a wide range of benchmarks (NL4Opt, Mamo ComplexLP, ComplexOR, IndustryOR, OptMATH) and compare it with diverse baselines, including both prompt-based methods (CoT, CoE, OptiMUS) and fine-tuned models (ORLM, etc.)

- I particularly appreciate that the authors went beyond automated metrics and conducted a manual inspection to measure the reliability of the verification agents . Validating the precision and recall of the verifiers adds significant credibility to the proposed multi-agent framework.

- The paper is well-written and easy to follow. The visualizations, particularly the framework overview (Figure 3) and the motivating examples (Figure 2), are high-quality and effectively communicate the core concepts and workflows.

**Weakness**

- Regarding the solution-side verification, the reliance on "toy instantiations" to verify logic seems like an unnecessary intermediate step. If the parameters are small enough for a toy instance, it might be more reliable to simply brute-force solve the problem to verify correctness, rather than relying on an LLM to interpret the solver's output.

- Furthermore, the fidelity of the toy instances is questionable; there is no guarantee that a simplified toy instance preserves the structural difficulty or the specific edge cases of the original large-scale problem. A model might work on the toy instance but fail on the full-scale data due to constraints that only activate under specific data distributions.

- While the dual-side verification targets different aspects (structure and solution), the underlying methodology for both is quite similar, that is, prompting an LLM to interpret and critique. This raises a question of whether these could be merged into a single, more robust verifier. A more rigid or formal verification method might be necessary to ensure true correctness, rather than stacking probabilistic LLM checks.

- The verification steps focus primarily on mathematical formulation correctness rather than the correctness of code implementation.

---

> ### Author Rebuttal · Authors · 2026-03-30
>
> ### **Weakness 1**
>
> We sincerely appreciate your insightful review.
>
> - **We do not use "toy instantiation" for small-scale problems**. For the problems in the standard dataset, Opt-Verifier directly solves the original problems to verify the output.
> - **The necessity of "toy instantiation" for large-scale problems**. In industrial scenarios (e.g., thousands of warehouses), solving the full-scale problem during the is computationally prohibitive. Toy instantiation provides a fast proxy to verify the modeling logic in seconds.
> - **Solution Interpretation and toy instantiation are two different steps.** In testing time, we cannot obtain the ground-truth solution. We interpret the solution from the original or the instantiated toy problems. Thus, we introduce the LLM-based solution interpretation step.
>
> ### **Weakness 2**
>
> We rely on two principles:
>
> *   The abstract mathematical and programmatic structure in the solver code (e.g., the formulation logic within a `for` loop) is fully agnostic to the problem scale. Verifying the abstract logic on a smaller $N$ validates the generalized mathematical structure.
> *   The framework samples **multiple groups of diverse toy data** (e.g., varying data distributions) to stress-test the abstract formulation.
>
> ### **Weakness 3**
>
> Rigid formal verification requires a formal ground truth to verify against, which is absent in our task. We separate the two verification steps to maintain system robustness.
>
> * The two verifiers process different modalities of information. The **structure-side verification** operates on the abstract mathematical formulation, while the **solution-side verification** processes the numerical solutions.
>
> * If merged, a single verifier would need to simultaneously process the natural language problem, the mathematical models, the solver code, and the numerical solver outputs. Assigning overly complex tasks to LLMs leads to severe context overload, resulting in hallucination.
>
>   ||Nl4Opt|Mamo Complex |ComplexOR | IndustryOR | OptMATH |
>   |-|-|-|-|-|-|
>   |Merged Single Verifier|92.0|60.7|78.9|41.0|31.9|
>   |Ours|96.5|66.7|78.9|45.0|34.3|
>
>
>
> ### **Question 2**
>
> Opt-Verifier does not simply stack LLM checks; rather, it introduces **additional external references or feedback**. Structure-side verification introduces the problem types as reference, while solution-side verification uses a solver to obtain numerical solutions as additional external feedback information.
>
> We first summarize the errors of LLMs (some are observed from [1]):
>
>   1. **Variables**: Incorrect definitions, omissions, redundant variables, and wrong variable types.
>   2. **Objective**: Wrong optimization direction, incorrect or missing objective terms.
>   3. **Constraints**: Incorrect, missing, or superfluous constraints, and confusing equalities with inequalities.
>   4. **Parameters**: Errors involving constant values, missing or wrong values.
>
>   We select some error cases made by LLM in OptMATH paired with the ground-truth model. We investigate the precision and recall on these examples (structure-side verification does not include parameters). The results demonstrate that the verification process is effective.
>
>   [1] Automated Optimization Modeling via a Localizable Error-Driven Perspective.
>
>   |Errors | Structure-Side Verification || Solution-Side Verification ||
>   | -| -| -| -| - |
>   | | Precision | Recall | Precision| Recall |
>   | Variable| 0.88| 0.83| 0.92| 0.95 |
>   | Constraint| 0.75| 0.67| 0.80| 0.79 |
>   | Objective| 0.94| 0.89| 0.97| 0.94 |
>   | Parameter|-|-| 0.89| 0.72 |
>
> ### **Weakness 4**
>
> Our method considers the correctness of code implementation, with a code-bug step when writing the solver code.
>
> ### **Question 1**
>
> We compare Opt-Verifier to standard test-time scaling approaches. The results show superiority in both effectiveness and token efficiency.
>
> ||NL4Opt||Mamo ComplexLP||IndustryOR||OptMATH||
> | -| - | - | -| -| -| -| -| -|
> ||Acc|Token| Acc| Token| ACC| Token| Acc | Token|
> |Majority Voting (3 models) |88.9 | 4957.7 | 50.2 | 7256.0 | 34.0 | 6632.5 | 20.5| 7913.8 |
> | Self-Correction (3 rounds) | 86.2 | 4483.2 | 46.9 | 6723.6 | 33.0 | 6264.9 | 17.5| 7260.1 |
> |Ours|96.5|5320.7|66.7| 7385.2 | 45.0 | 6819.8 | 34.3| 8251.7 |
>
> ### **Limitations**
>
> We will cite the works in our revised paper and try our best to implement them for benchmarking. Existing works fall into:
>
> - **Hint-based Enhancement (OptiMind, AlphaOpt)**: These methods collect external hints or experiences to enhance the modeling process.
>
> - **Validation and Feedback Mechanisms**:
>
>   - **Code Debugging Feedback (OptiMUS):** These systems trigger corrections based on solver errors.
>   - **Standard Self-Correction (LLMOPT, OptiMind):** These methods rely on the LLM's internal reasoning to verify mathematical formulations.
>
>   - **Our Framework**. Our work designs new **correction feedback based on systematic mathematical and logical checks**, which is fundamentally different from the other categories.

---

> > ### Author Rebuttal · Reviewer_TC8b · 2026-04-03
> >
> > Thank you to the authors for their detailed response and the additional empirical results. After reviewing the rebuttal and the comments from the other reviewers, I find that while the authors made a commendable effort to clarify their approach, several of my core concerns remain unresolved. Therefore, I believe it is appropriate to maintain my current score of 3.
> >
> > Specifically, my remaining reservations are the following:
> >
> > The authors argue that toy instantiation provides a fast proxy because solving **full-scale industrial problems** is computationally prohibitive. However, how do you ensure that the LLM captures the actual problem structure (i.e., complicated auxiliary constraints or specific edge-case distributions) rather than simply rewriting a small, standard instance? It would be nice if the author have a case study for running the pipeline on an industrial scale problem.
> >
> > The authors state that they introduce "external references" like the distilled problem type to avoid simply **stacking LLM checks**. However, if this distillation is performed by the same base LLM using the exact same input information, it risks amplifying the LLM's inherent biases. If the model misinterprets a subtle constraint during the initial distillation, the verifier will likely just confirm its own flawed logic, leading to confirmation bias rather than objective verification. Furthermore, if the information and model are identical, it is unclear why this logic cannot be verified accurately in a single round.
> >
> > The authors' new table shows a 0.67 **recall rate** for constraint errors during structure-side verification. Given that constraints are often the most critical and complex part of OR formulations, missing a third of constraint errors reinforces my concern that stacking probabilistic LLM checks is not a substitute for more rigid or formal verification.Lack of Direct Empirical Benchmarks: While I appreciate the authors categorizing concurrent works like LLMOPT, OptiMind, and AlphaOpt, conceptual categorization is not a substitute for head-to-head empirical benchmarking. Without comparing Opt-Verifier against these highly relevant iterative refinement methods, the state-of-the-art claims remain difficult to fully validate.
> >
> > Overall, while the paper presents an interesting conceptual direction, the methodology requires more rigorous grounding to overcome the inherent probabilistic weaknesses of LLM-as-a-judge approaches.

---

> > > ### Author Response · Authors · 2026-04-04
> > >
> > > We sincerely thank you for your constructive follow-up. We address your three remaining reservations below:
> > >
> > > - We evaluated Opt-Verifier on a **real-world AC power grid unit commitment problem** involving over 1,000,000 variables. To handle this computationally prohibitive scale, we abstract the problem by replacing the massive explicit parameters with symbolic representations. Our tests confirm that Opt-Verifier **successfully modeled this complex task**. We provide a concise snippet of the generated Gurobi code.
> > >
> > >   > (Simplified Version Due to the Limited Space) A regional power system, comprising $N$ generating units, $L$ transmission lines, and $B$ buses, requires a generation  schedule spanning $T$ consecutive time periods. The operational objective is to minimize the total cost, which includes the cumulative fuel costs—quadratic functions of real power output for each generating unit with coefficients $a_i$, $b_i$, and $c_i$—along with the start-up costs $SUC_i$ and shut-down costs $SDC_i$.
> > >   >
> > >   > For each time period, the total real power injected into the system by all online generators must equal the sum of the total real power demand across all loads, $\sum_b D^P_{b,t}$, and the total real power losses. At every bus, the net injection of reactive power—comprising the reactive power output minus the local reactive load demand $D^Q_{b,t}$—must be balanced to zero.
> > >   >
> > >   > When a unit is committed online, its real power output must be in its minimum $P_i^{\min}$ and maximum $P_i^{\max}$ stable generation limits, and its reactive power output must be kept within minimum $Q_i^{\min}$ and maximum $Q_i^{\max}$ reactive capability bounds. The combined real and reactive power output of the unit must always reside within $S_i^{\max}$.
> > >   >
> > >   > The increase in output from one period to the next cannot exceed the ramp-up rate $RU_i$, and the decrease ramp-down rate $RD_i$. Once a unit is started up, it must remain online for a minimum duration of $T_i^{on}$ periods, and once shut down, it must remain offline duration of $T_i^{off}$ periods.
> > >   >
> > >   > ...
> > >   ```
> > >   def build_ac_unit_commitment_model(N, L, B, T, params):
> > >     """
> > >     N: Number of generators
> > >     L: Number of transmission lines
> > >     B: Number of buses
> > >     T: Number of time periods
> > >     params: Dictionary containing all symbolic parameters (e.g., limits, costs, network data)
> > >     """
> > >     # 1. Decision Variables
> > >     # Commitment status
> > >     u = model.addVars(N, T, vtype=GRB.BINARY, name="u")
> > >     ...
> > >
> > >     # 2. Constraints
> > >     for t in range(T):
> > >         # 2.1 Nodal Power Balance (Active and Reactive)
> > >         for b in range(B):
> > >             # Active power balance: Generation - Load = Net Flow Out
> > >               model.addConstr(
> > >                   gp.quicksum(P[i,t] for i in gens_at_b) - params['D_P'][b,t] ==
> > >                   gp.quicksum(FP[l,t] for l in lines_from_b) - gp.quicksum(FP[l,t] for l in lines_to_b),
> > >                   name=f"P_Balance_b{b}_t{t}"
> > >               )
> > >             ...
> > >   ```
> > >
> > > - **Regarding external references**. This touches on a core observation in our paper (detailed in Example, Section 3). We found that **LLMs possess the necessary OR knowledge, but often fail to utilize it for multi-hop implicit reasoning during a single generative pass.**
> > >
> > >   - In a single round, the LLM is overwhelmed by translating text, mapping variables, and writing constraints simultaneously. Consequently, it frequently misses implicit constraints like "flow balance". However, if we simply ask the LLM to classify the problem first, it outputs "Maximum Flow"—this specific keyword acts as a trigger, successfully activating its latent OR knowledge.
> > >   - We explicitly **do not** ask the verifier, "Is your previous model correct?" (which would indeed trigger confirmation bias). Instead, our Distillation Agent independently extracts a standard "Structure Schema" purely from the text. The Evaluation Agent then performs an objective alignment check: "Does the generated model contain the components required by this standard Schema?" Because the reference is built independently by deeply exploiting the latent knowledge of LLM, it is not the self-justification loop.
> > >
> > > - We evaluated Opt-Verifier alongside LLMOPT, AlphaOpt, and OptiMind. Furthermore, we integrated Opt-Verifier's verification module into these baselines.  As shown in the table below, our standalone framework achieves superior performance. More importantly, **Opt-Verifier serves as a highly effective enhancement**. We will include these results in the revised manuscript.
> > >
> > >   | Method| NL4Opt | Mamo ComplexLP | IndustryOR | OptMATH |
> > >   | - | -| -| -| -|
> > >   | **Ours (GPT4o-mini)** |96.5|66.7| 45.0| 34.3|
> > >   | LLMOPT| 80.3   | 44.1| 29.0| 12.5|
> > >   | **LLMOPT+Ours**| 85.1| 54.5| 40.0| 31.3|
> > >   | AlphaOpt (GPT4o-mini) | 90.3| 62.6 | 38.0| 31.9|
> > >   | **AlphaOpt+Ours**| 93.4| 68.7| 44.0| 34.3|
> > >   | OptiMind| 94.1| 64.0| 42.0| 33.7|
> > >   | **OptiMind+Ours**| 96.2| 69.7| 48.0| 34.9|

---

### Official Review · Reviewer_dGsx · 2026-03-11

**Soundness:** 2
**Presentation:** 2
**Significance:** 2
**Originality:** 2
**Overall Recommendation:** 4
**Confidence:** 4

**Summary:**

This paper proposes a new multi-agent cooperation system for optimization modeling. The method verifies generated models from both the structure-side and the solution-side. Experiments on several OR benchmarks show consistent improvements over CoE, OptiMUS, and ORLM.

**Compliance With Llm Reviewing Policy:**

Affirmed.

**Final Justification:**

The rebuttal period addressed my main concerns about this paper.

**Key Questions For Authors:**

1. Do the authors conduct a scaled evaluation of verifier quality itself beyond pure final accuracy?
2. Can the authors provide confidence interval in future editions of the paper?
3. Could the authors provide some explanation for the 19 instances chosen from ComplexOR?

**Limitations:**

yes

**Strengths And Weaknesses:**

Strengths:
1. The dual-side verification idea is easy to follow.
2. The emperical results are good for Opt-Verifier. This method outperforms baseline includes CoE, OptiMUS, and ORLM.

Weaknesses:
1. Both structure-side and solution-side checks rely on LLMs-as-a-judge. The paper did not fully address the reliability at scale.
2. For the experiment the scale is relatively small without confidence interval. 19 problem instances are chosen from ComplexOR without justification for instance selection.

---

> ### Author Rebuttal · Authors · 2026-03-30
>
> ### **Weakness 1 and Question 1**
>
> We sincerely thank you for your review. We would like to respectfully clarify that we have studied and addressed the reliability of our verifiers. To further test the reliability of the verification process, we first summarize the errors of LLMs (some are observed from [1]):
>
> 1. **Variables**: Mistakes related to decision variables, including incorrect definitions, omissions, redundant variables, and wrong variable types.
> 2. **Objective**: Errors in the objective function, such as wrong optimization direction, incorrect or missing objective terms, and failures in applying advanced modeling techniques.
> 3. **Constraints**: Issues with problem constraints, including incorrect, missing, or superfluous constraints, confusing equalities with inequalities, and improper advanced constraint modeling.
> 4. **Parameters**: Errors involving constant values, primarily incorrect parameter definitions (missing or wrong values), and parameter misuse (wrong units or scales).
>
> We select some error cases in OptMATH paired with the ground-truth model. We inspect the LLM-generated optimization models and investigate the precision and recall on these examples (structure-side verification does not include parameters). The results demonstrate that the verification process is effective for the LLM-generated errors.
>
> [1] Automated Optimization Modeling via a Localizable Error-Driven Perspective.
>
> **Table 1**: Reliability of verifiers on the LLM-generated errors.
>
> |                   | Structure-Side Verification |        | Solution-Side Verification |        |
> | ----------------- | --------------------------- | ------ | -------------------------- | ------ |
> |                   | Precision                   | Recall | Precision                  | Recall |
> | Variable Errors   | 0.88                        | 0.83   | 0.92                       | 0.95   |
> | Constraint Errors | 0.75                        | 0.67   | 0.80                       | 0.79   |
> | Objective Errors  | 0.94                        | 0.89   | 0.97                       | 0.94   |
> | Parameter Errors  | -                           | -      | 0.89                       | 0.72   |
>
> ### **Weakness 2 and Question 3**
>
> We sincerely appreciate your constructive feedback regarding the experimental scale and the selection of benchmark instances.
>
> - **Dataset Scale for the Experiments**. The datasets we used in this paper include NL4Opt, Mamo ComplexLP, ComplexOR, IndustryOR, and OptMATH. Except for ComplexOR, the other datasets contain a large amount of data with more than 100 instances. The NL4Opt dataset has 289 problems, Mamo ComplexLP has 211, IndustryOR has 100, and OptMATH has 166. The problems cover scenarios across more than 20 fields, where the data scale is not small.
>
> - **Explanation for the 19 ComplexOR Instances**. We followed the exact experimental protocol and data established by **LLMOPT** [1], which uses the ComplexOR dataset with 19 instances.
>
> - **Updating Experiments with Refined Datasets**. We have updated our experimental evaluation to include the latest and cleaned versions of the benchmark datasets (the recently released, well-recognized comprehensive LLM4OR benchmark [2]). We also test our method on the cleaned 18-instance ComplexOR released in LLM4OR. This dataset is used in the paper [2,3].
>
>   [1] LLMOPT: Learning to Define and Solve General Optimization Problems from Scratch.
>
>   [2] A Survey of Optimization Modeling Meets LLMs: Progress and Future Directions.
>
>   [3] StepORLM: A Self-Evolving Framework With Generative Process Supervision For Operations Research Language Models.
>
>   **Table 2**. Experiments on the revised ComplexOR.
>
>   |                    | Standard | CoT  | ORLM | CAFA | CoE  | OptiMUS | Ours |
>   | ------------------ | -------- | ---- | ---- | ---- | ---- | ------- | ---- |
>   | ComplexOR (LLM4OR) | 42.9     | 39.2 | 50.0 | 46.4 | 57.1 | 46.8    | 61.1 |
>
> ### **Question 2**
>
> Yes, we will provide confidence intervals in the revised edition of the paper. To address the stochastic nature of LLMs and rigorously prove the stability of our method, we will report the standard deviation on 5 independent trials.
>
> **Table 3**: The statistics on standard deviation.
>
> |          | NL4Opt      | Mamo ComplexLP | ComplexOR   | IndustryOR | OptMATH    |
> | -------- | ----------- | -------------- | ----------- | ---------- | ---------- |
> | Standard | 64.6 (0.5)  | 27.9  (1.3)    | 31.5 (1.3)  | 24.0 (0.9) | 15.6 (0.4) |
> | CoT      | 69.3  (0.3) | 34.5  (1.6)    | 36.8 (2.2)  | 27.0 (1.1) | 18.6 (0.2) |
> | CoE      | 71.3  (0.6) | 44.5  (0.9)    | 68.4 (1.6)  | 29.0 (1.3) | 19.8 (0.5) |
> | OptiMUS  | 83.0  (0.6) | 45.0 (1.1)     | 73.6 (2.5)  | 31.0 (0.8) | 20.2 (0.5) |
> | Ours     | 96.5  (0.4) | 66.7 (1.1)     | 78.9 (0.00) | 45.0 (1.2) | 34.3 (0.4) |

---

> > ### Author Rebuttal · Reviewer_dGsx · 2026-04-02
> >
> > Thank you for your detailed response. I'll maintain my positive score 4.

---

> > > ### Author Response · Authors · 2026-04-04
> > >
> > > Thank you very much for your positive feedback. We are glad that our clarifications and additional empirical results have addressed your concerns, and we will ensure all these details are fully incorporated into the final manuscript.

---

### Official Review · Reviewer_kyEe · 2026-03-13

**Soundness:** 2
**Presentation:** 3
**Significance:** 2
**Originality:** 2
**Overall Recommendation:** 4
**Confidence:** 3

**Summary:**

This paper studies LLM-based optimization modeling from natural language descriptions. It proposes Opt-Verifier, a dual-side verification framework that refines generated models using two signals: structure-side verification and solution-side verification. Experiments on several optimization-modeling benchmarks show improved solving accuracy over multiple baselines.

**Compliance With Llm Reviewing Policy:**

Affirmed.

**Final Justification:**

The authors’ rebuttal has clarified the motivation and provided more evidence, which addressed my main concerns.

**Key Questions For Authors:**

1. The paper positions itself against prior work by emphasizing dual-side verification. Could the authors clarify more precisely how Opt-Verifier differs from prior optimization-modeling systems with richer validation and feedback mechanisms, such as OptiMUS, Step-Opt, SIRL, and StepORLM?
2. Could the authors provide a more controlled comparison on a unified benchmark setting?
3. How representative are the negative samples in the “Quantity Analysis of Verifications” section of real LLM-generated modeling errors? Could the authors provide an analysis of real LLM-generated modeling errors?

**Limitations:**

The setup in the “Quantity Analysis of Verifications” section is somewhat limited. The negative samples are created by “randomly deleting or rewriting some of the variables and constraints,” which may not fully match the kinds of errors that LLMs make in optimization modeling. The paper should discuss this limitation more clearly.

**Strengths And Weaknesses:**

Strengths:
1. The paper is well written and easy to follow. The method section clearly explains the roles of different agents and the overall verification–feedback–refinement process.
2. The proposed framework is clear and well-structured. The motivation and role of each component are generally easy to follow, and the overall method is organized in a logical way.
3. The empirical study is comprehensive. The paper includes component ablations, structure-level ablations, and additional analyses on efficiency and different model settings.

Weakness:
1. The novelty is incremental. While the framework is well organized, its core ideas build on directions that have already been explored in prior work, including richer validation and feedback mechanisms in optimization modeling systems such as OptiMUS, Step-Opt, SIRL, and StepORLM. As a result, the contribution seems to lie more in combining these components into a unified framework.
2. The empirical comparison is not fully fair. The main results table combines the authors’ own results with numbers reported in other papers. Also, different prior works use different cleaned benchmark versions and subsets, so these results are not directly comparable. This is especially important here because recent work has already shown that several optimization-modeling benchmarks contain non-trivial errors, and has released cleaned versions and consolidated results, including the LLM4OR benchmark [1] and the revised benchmark released by SIRL [2]. Therefore, a unified evaluation on the same cleaned benchmark version would be much more convincing.
[1] Xiao Z, Xie J, Xu L, et al. A survey of optimization modeling meets llms: Progress and future directions. arXiv preprint arXiv:2508.10047, 2025.
[2] Chen Y, Xia J, Shao S, et al. Solver-informed RL: Grounding large language models for authentic optimization modeling. arXiv preprint arXiv:2505.11792, 2025.

---

> ### Author Rebuttal · Authors · 2026-03-30
>
> ### **Weaknesses 1 and Question 1**
>
> We sincerely thank you for your constructive feedback. We respectfully clarify that Opt-Verifier is not a mere combination of existing components, but introduces a fundamentally different design of feedback paradigm.
>
> - **Classification of existing works**. Existing works with feedback and validation machemisms primarily fall into two categories:
>   - **Code Debugging feedback (OptiMUS):** These systems trigger corrections primarily based on solver error messages. If a generated model is logically or mathemactically flawed but the solver executes successfully without bugs, these systems fail to detect the error. The feedback is sparse.
>   - **Standard Self-Correction (Step-ORLM, LLMOPT):** These methods rely on the LLM's intrinsic reasoning to verify mathematical formulations. The validation task is still challenging and this approach frequently suffers from hallucinations, as the LLM struggles to identify its own logical blind spots.
> - **Our framwork**. In contrast, our work design **dense correction feedback based on systematic mathematical and logical check**, which is fundamentally different from both categories. Opt-Verifier does not simply stack internal probabilistic LLM checks; rather, it introduces additional external reference or feedback information.
>   *   Structure-side verification intruduce the problem types as reference to provide a structural correctness feedback.
>   *   Solution-side verification uses solver to obtain numerical solutions as additional external feedback information.
>
>
> - **Why we need such a framework**. Please refer to our response to Reviewer 77UY, where we discuss the necessity of our verification framework. Simple verification machanisms fail to capture the intrinsic errors in the optimization models. Furthermore, we conduct an ablation studies with code-based verification and self-correction framework as follows.
>
>   **Table 1**:  Comparison between the existing feedback and correction paradigms.
>
>   | Method| NL4Opt | Mamo ComplexLP | ComplexOR | IndustryOR | OptMATH |
>   | -| -| -| -| -| -|
>   | Code-based Validation | 82.7| 42.6| 42.1| 28.0| 16.2    |
>   | Self-Correction| 86.2   | 46.9| 63.1| 33.0| 17.5|
>   | Ours| 96.5   | 66.7| 78.9| 45.0| 34.3|
>
> ### **Weakness and Question 2**
>
> To ensure a strictly fair and controlled comparison, we are adopting the latest cleaned benchmark versions from LLM4OR to conduct a unified evaluation. The results are presented in Table 2 below.
>
> **Table 2**: Comparison on the cleaned benchmarks.
>
> |          | NL4Opt | Mamo ComplexLP | ComplexOR | IndustryOR | ReSocratic |
> | -| -| -| -| -| -|
> | Standard | 61.2   | 57.7| 42.9      | 38.1       | 48.4       |
> | CoT      | 62.2 | 42.3| 39.2| 40.5 | 43.6 |
> | ORLM | 73.8 | 59.5| 50.0| 42.9 | 61.8 |
> | CAFA | 68.1 | 44.5 | 46.4| 41.1 | 40.1 |
> | CoE| 66.7 | 50.6 | 57.1| 31.2 | 71.2 |
> | OptiMUS| 76.2 | 46.8 | 46.8| 45.2 | 87.6 |
> | Ours | 86.0 | 78.3 | 61.1| 52.3 | 89.6 |
>
> ### **Question 3 and Limitation**
>
> We sincerely thank you for pointing out this important limitation regarding the representativeness of our negative samples. Regarding your request for an analysis of real LLM-generated modeling errors, our framework's design is fundamentally motivated by these exact real-world errors.
>
> - **LLM-Generated Errors**. We first summarize the errors of LLMs (some are observed from [1]):
>
>   1. **Variables**: Mistakes related to decision variables, including incorrect definitions, omissions, redundant variables, and wrong variable types.
>   2. **Objective**: Errors in the objective function, such as wrong optimization direction, incorrect or missing objective terms, and failures in applying advanced modeling techniques.
>   3. **Constraints**: Issues with problem constraints, including incorrect, missing, or superfluous constraints, confusing equalities with inequalities, and improper advanced constraint modeling.
>   4. **Parameters**: Errors involving constant values, primarily incorrect parameter definitions (missing or wrong values) and parameter misuse (wrong units or scales).
>
>   [1] Automated Optimization Modeling via a Localizable Error-Driven Perspective.
>
> - **Empirical Results**. We select some error cases in OptMATH paired with the ground-truth model. We inspect the LLM-generated optimization models and investigate the precision and recall on these examples (structure-side verification does not include parameters). The results demonstrate that the verification process is effective for the LLM-generated errors.
>
>   **Table 3**: Reliability of verifiers on the LLM-generated errors.
>
>   |  | Structure-Side Verification || Solution-Side Verification ||
>   |-|-|-|-|-|
>   |  | Precision| Recall | Precision| Recall |
>   | Variable Errors   | 0.88| 0.83| 0.92| 0.95   |
>   | Constraint Errors | 0.75| 0.67   | 0.80| 0.79   |
>   | Objective Errors  | 0.94| 0.89   | 0.97| 0.94   |
>   | Parameter Errors  | -| -  | 0.89  | 0.72   |

---

> > ### Author Rebuttal · Reviewer_kyEe · 2026-04-04
> >
> > Thank you for the rebuttal and the additional empirical results. I appreciate the authors’ efforts to clarify the motivation and provide more evidence.

---

> > > ### Author Response · Authors · 2026-04-04
> > >
> > > We sincerely thank you for taking the time to review our rebuttal and for your continued support of our work. We are very glad to hear that our detailed responses and the additional empirical results have addressed your concerns.

---

### Official Review · Reviewer_77UY · 2026-03-15

**Soundness:** 3
**Presentation:** 3
**Significance:** 3
**Originality:** 3
**Overall Recommendation:** 4
**Confidence:** 2

**Summary:**

This paper studies the problem of automatically generating optimization models and solver code from natural language descriptions. The authors observe that existing LLM-based approaches often generate incorrect optimization formulations, such as missing constraints or inconsistent modeling assumptions. These errors may not always be detected by solvers and can lead to incorrect solutions even when the code executes successfully.

To address this issue, the paper proposes Opt-Verifier, a multi-stage framework that incorporates structure analysis and verification mechanisms into the modeling pipeline. The framework first extracts multi-level structural information from the problem description, then generates an optimization model, and finally applies both structure-side and solution-side verification to detect modeling errors and refine the generated formulation. Experimental results on several benchmark datasets show consistent improvements in solving accuracy compared to several baselines.

**Compliance With Llm Reviewing Policy:**

Affirmed.

**Key Questions For Authors:**

The main concern of this paper is the necessity of the proposed multi-stage framework. The proposed system introduces a relatively complex pipeline. While the design is intuitive, the paper does not clearly demonstrate that such a complex framework is necessary compared to simpler alternatives.

For example, a simple verification idea is like:

1. Use a strong general-purpose LLM to translate the natural language problem into a structured mathematical formulation.

2. Show this formulation to an optimization-oriented LLM to generate the corresponding optimization code.

3. Come back to the strong LLM and verify whether all constraints and problem requirements are satisfied.

Such a pipeline may already capture many of the benefits of the proposed framework without requiring multiple specialized agents. Additionally, I tested two complexOR problems (h3,h4) in this paper in the default ChatGPT without logging-in by directly asking the math formulation of the given problem, and the model was able to produce reasonable mathematical formulations for these problems. While this is only anecdotal evidence, it suggests that strong modern LLMs may already handle some of these tasks reasonably well with simpler prompting pipelines. I have kept the prompts and responses and can provide them if needed.

Therefore, it would significantly strengthen the paper if the authors could provide examples of problems that cannot be reliably solved by strong general-purpose LLMs but can be solved by the proposed framework. Demonstrating such cases would help clarify the necessity and practical advantage of the proposed multi-stage design.

**Strengths And Weaknesses:**

Strengths

S1: The problem itself is important and impactful. Automatically translating natural language descriptions into correct optimization models is an important problem for both the optimization and machine learning communities. Improving the reliability of such pipelines could have practical value for real-world applications.

S2: The proposed method is intuitive and well-motivated.

S3: The empirical analyses are very thorough and impressive.

Weakness

W1: The necessity of the proposed complex framework is not clearly discussed. The paper introduces a relatively complex multi-stage pipeline, but it does not clearly demonstrate whether such complexity is necessary compared to simpler alternatives. See my question below for more details.

---

> ### Author Rebuttal · Authors · 2026-03-30
>
> Below, we address your core concern regarding the necessity of the proposed Opt-Verifier compared to simpler general-purpose LLM pipelines.
>
> - **Concrete Example**. While strong general-purpose LLMs demonstrate capable zero-shot and self-correction abilities for standard optimization problems, they exhibit systematic blind spots—specifically confirmation bias—when handling complex modeling tasks with implicit constraints. Consider the following example,
>
>   ```
>   A certain factory needs to use a special tool for over $n$ planning stages. At stage $j$, $r_j$ specialized tools are needed. At the end of this stage, all tools used within this stage must be sent for repair before they can be reused. There are two repair methods: one is slow repair, which is cheaper (costs $b$ per tool) but takes longer ($p$ stages to return); the other is fast repair, which costs $c$ per tool $(c > b)$ and is faster, requiring only $q$ stages to return $(q < p)$. If the repaired tools cannot meet the needs, new ones must be purchased, with a cost of $a$ per new tool $(a > c)$. This special tool will no longer be used after $n$ stages. Determine an optimal plan for purchasing and repairing the tools to minimize the cost spent on tools during the planning period.
>   n = 10 # number of stages
>   r = [3, 5, 2, 4, 6, 5, 4, 3, 2, 1] # tool requirements per stage, indexing starts at 1
>   a = 10 # cost of buying a new tool
>   b = 1 # cost of slow repair
>   c = 3 # cost of fast repair
>   p = 3 # slow repair duration
>   q = 1 # fast repair duration
>   ```
>
>   **Code generated by ChatGPT (using a simple workflow):**
>
>   ```
>   # -----------------------------
>   # Decision Variables
>   # -----------------------------
>   x = m.addVars(n, vtype=GRB.INTEGER, name="New")    # new tools purchased
>   y = m.addVars(n, vtype=GRB.INTEGER, name="Slow")   # tools sent to slow repair
>   z = m.addVars(n, vtype=GRB.INTEGER, name="Fast")   # tools sent to fast repair
>   s = m.addVars(n, vtype=GRB.INTEGER, name="Available")  # tools available at stage
>
>   # -----------------------------
>   # Constraints
>   # -----------------------------
>   for j in range(n):
>       # Tools returned from slow repair
>       slow_return = y[j - p] if j - p >= 0 else 0
>       # Tools returned from fast repair
>       fast_return = z[j - q] if j - q >= 0 else 0
>
>        # Error 1: Fails to carry over unused tools from inventory
>       m.addConstr(s[j] == x[j] + slow_return + fast_return, name=f"Availability_{j}")
>
>       m.addConstr(s[j] >= r[j], name=f"Requirement_{j}")
>
>       # Error 2: Forces all used tools to be repaired immediately
>       m.addConstr(y[j] + z[j] == r[j], name=f"Repair_Balance_{j}")
>
>   # -----------------------------
>   # Objective: minimize total cost
>   # -----------------------------
>   m.setObjective(
>       sum(a*x[j] + b*y[j] + c*z[j] for j in range(n)),
>       GRB.MINIMIZE
>   )
>   ```
>
>   **Why the simple pipeline fails**. The generated model runs perfectly and outputs a numerical result. A standard LLM self-correction pipeline, which relies on reading the code and checking for execution errors, typically accepts this model because it superficially matches the text. However, it completely misses the **implicit inventory flow balance** (i.e., holding unused tools, or holding used tools to repair later).
>
>   **Our Opt-Verifier framework can identify the errors and provide the correct answer**.
>
>   - **Structure-Side Verification**. By abstracting the model back into an OR structure, the agent compares it against the standard inventory problem structure, immediately identifying the missing state-transition constraints for tool inventories.
>   - **Solution-Side Verification**. By solving the problem, the agent translates the numerical output into natural language (e.g., "3 tools were purchased yesterday and only 1 was used, but today the available tools dropped to 0"). This exposes a physical and logical contradiction that is invisible to a standard code-checking pipeline.
>
> - **Experiment Results**. We compare our method with two baselines. One is a simple workflow with a weaker LLM (GPT4o-mini), the other is a simple workflow with a stronger LLM (GPT-4o). The results are provided as follows. In the experimental results, we find that our workflow can enhance the modeling ability of weaker LLMs and even beat strong LLMs with simple workflows.
>
>   | Method                        | NL4Opt | Mamo ComplexLP | ComplexOR | IndustryOR | OptMATH |
>   | ----------------------------- | ------ | -------------- | --------- | ---------- | ------- |
>   | Simple Framework (GPT-4o)     | 96.3   | 54.5           | 73.6      | 39.0       | 26.5    |
>   | Simple Framework (GPT4o-mini) | 76.1   | 47.8           | 47.3      | 29.0       | 19.9    |
>   | Ours (GPT4o-mini)             | 96.5   | 66.7           | 78.9      | 45.0       | 34.3    |

---

> > ### Author Rebuttal · Reviewer_77UY · 2026-04-07
> >
> > I appreciate the authors for providing this example. When dropping this context into the free version of ChatGPT, I can recover the author's results. That is, the output of ChatGPT has the listed two errors. It would be great to include the example in the paper.

---

### Decision · Program_Chairs · 2026-04-30

**Decision:**

Accept (regular)

**Comment:**

This paper tackles the challenging of synthesizing optimization models from natural language description, using LLMs. In particular, to address failure modes in prior approaches, the submission proposes a multi-agent "dual-side verification" architecture, which leverages LLM natural language reasoning abilities to empirically verify (a) the structure of the generated optimization model and (b) the numerical solutions on smaller instances, that the solution makes sense in the natural language description of the task.

Reviewers generally appreciated the solid empirical improvements, and a reviewer particularly noted that the authors did extensive manual checking instead of just fully automating the evaluation. The writing quality of the paper is also well-appreciated.

I recommend acceptance.

The authors are encouraged to revise the submission based on the discussion, for example, motivating the more-complicated architecture as opposed to the simpler ones reviewers asked about.